

# Simulation analysis of 3D stability of a landslide with a locking segment: A case study of Tizicao landslide in Maoxian County, Southwest China

Yuntao Zhou[1,3], Xiaoyan Zhao[1]*, Guangze Zhang[2], Bernd Wünnemann[1], Jiajia Zhang[3], Minghui Meng[4]

[1]Department of Geology Engineering, Faculty of Geosciences and Environmental Engineering, Southwest Jiaotong University, Chengdu, 611756, Sichuan, China

[2]China Railway Eryuan Engineering Group Co., Ltd., Chengdu, 610031, Sichuan, China

[3]Institute of Exploration Technology, Chinese Academy of Geological Sciences, Chengdu, 611734, Sichuan, China

[4]Sichuan Huadi Construction Engineering Co., Ltd., Chengdu, 610081, Sichuan, China

*Correspondence*: Xiaoyan Zhao (xyzhao2@swjtu.edu.cn)

**Abstract.** Rock bridges, also known as locking rock masses in landslides, influence the three-dimensional (3D) stability and deformation patterns of landslides. However, it is always difficult to simulate rock bridges with continuous grid models in three-dimensional landslides due to their discontinuous deformation. Tizicao landslide, located in Maoxian County, Southwestern China, is a typical landslide with a super-large rock mass volume of about $1,388.2 \times 10^4$ m$^3$ and characterized by a locking segment. To explore a better rock bridge model to simulate 3D stability and deformation behaviors of the Tizicao landslide, three rock bridge models were introduced into the FLAC3D program, including the intact rock mass model (IRMM), the Jennings model (JM), and the contact surface model with high strength parameters (CSM-HSP). The CSM-HSP model was eventually used in the FLAC3D program to obtain the 3D deformation characteristics of the landslide. Also, the two-dimensional (2D) stability of the Tizicao landslide was analyzed using the GeoStudio program. The simulation results indicate that the Tizicao landslide is stable overall under the current conditions, owing to the existence of the locking segment in the southern front, which is consistent with the field deformations and monitoring data. The overall stability and the local deformation of the landslide are found to be influenced by the locking segment when compared to 2D and 3D stability. There is a linear relationship between the locking ratio and the factor of safety (Fos) for the 2D landslide with a locking segment, while an approximate quadratic parabola for the 3D stability. Finally, the laws of the 3D Fos varying with locking ratios and strength parameters of the locking masses and the sliding surface were analyzed. The advantages and disadvantages of the three rock bridge models in simulating the 3D stability of landslides with a locking segment were also discussed.

## 1. Introduction

A landslide with a locking segment is a geological phenomenon in which a locking segment exists along the sliding surface and the critical failure is controlled by the shear properties of the locking mass (Xu et al., 2010; Huang, 2012; Lin et al., 2018). A landslide of this type usually holds huge potential energy (Huang, 2012). Once the locking masses are cut off, the accumulated energy would suddenly be released and a mass of fragmental materials from the landslide would affect the residential areas and infrastructures below the landslide, thus frequently resulting in catastrophic effects and severe casualties (Yin et al., 2011; Lin et al., 2018; Wang et al., 2019). The locking masses serve as a key in analyzing the stability of a landslide with a locking segment. However, uncertain positions, irregular shapes, and certain varying curvatures of a locking segment in the landslide make a 2D stability analysis more difficult. Meanwhile, 2D stability analysis is often applied to engineering reinforcement design and is relatively conservative. The results obtained through the 2D stability analysis can



only represent the local stability of landslides (Li et al., 2010; Park et al., 2017). Therefore, 3D stability methods play a major role in assessing and predicting the overall stability of the landslide with a locking segment.

At present, the commonly used 3D stability methods of landslides include limit equilibrium methods and numerical simulation methods. To account for the 3D stability of slopes, many 3D limit equilibrium methods have been proposed (Hovland, 1977; Leshchinsky et al., 1986; Hungr et al., 1989; Lam and Fredlund, 1993). Most of these methods are simply based on the extension of the 2D limit equilibrium slice methods proposed by Bishop (1955), Morgenstern and Price (1965), or Spencer (1967). The inherent limitations of the deformation and failure mode analysis remain in these 3D solutions. Fortunately, the simulation methods can be used not only to provide a simple and useful way of analyzing the 3D stability but also to analyze the deformation and failure tendency of landslides. 3D simulation methods have been employed to determine the 3D stability of slopes/landslides (Deng et al., 2011; Wang et al., 2013; Zhang et al., 2013; Ma et al., 2020). Nevertheless, the rock bridge effect is not considered in most numerical simulations of 3D stability analysis of landslides. As indicated by 2D or 3D planar stability simulation studies of slopes/landslides with rock bridges, rock bridges determine the stability and control the failure mode of the slopes/landslides (Stead et al., 2006; Huang et al., 2015; Glueer and Loew, 2015). In addition, several problems related to rock bridges are yet to be solved, including the response of the 3D landslide stability to rock bridges, the control of rock bridges over slope deformation, and the realization of rock bridges in numerical simulation software.

Some researchers have found that the stability of slopes/landslides with rock bridges is closely related to the length, penetration rate, strength parameters, joint strength parameters, relative positions (direction, coplane, or non-coplane), and shape of rock bridges (Einstein et al., 1983; Tuckey and Stead, 2016; Romer and Ferentinou, 2019; Zhang et al., 2020). The qualitative relationships between the 2D slope/landslide stability and these parameters are generated. However, there is a lack of in-depth quantitative study on the relationships, especially on the 3D slopes/landslides stability.

The objective of this work is to present a better rock bridge model to simulate 3D stability and deformation behaviors of the Tizicao landslide. A large number of field investigation data and monitoring data show that this landslide is a typical landslide with a locking segment, and the locking segment controls the 3D stability of the landslide (Zhou et al., 2022). Then three rock bridge models were introduced into the FLAC3D program, including the intact rock mass model (IRMM), the Jennings model (JM), and the contact surface model with high strength parameters (CSM-HSP). Meanwhile, the analysis results of 3D stability were compared with those of 2D stability obtained using the GeoStudio program to analyze the differences between the overall stability and the local stability of the landslide. In addition, to explore the effects of the locking rock masses on 3D stability, the laws of the 3D Fos varying with locking ratios and strength parameters of the locking masses and sliding surface were analyzed. The comparative analysis of the three rock bridge models in simulating the 3D stability of landslides with a locking segment wass further discussed to analyze their advantages and disadvantages.

**2. Study site**

The Tizicao landslide is located in Maoxian County, Sichuan Province, Southwestern China (Fig. 1), with geographical coordinates of 31°53′14.89″N and 103°40′51.12″E. It lies on the right bank of the Minjiang River and faces Shidaguan Town on the left bank of the river (Fig. 2a). The Tizicao landslide has a length of about 680 m, a width of 570 m (Fig. 2a), an average thickness of about 39.1 m, and a volume of 1,388.2 × 10⁴ m³. The landslide has a huge gravitational potential energy since the relative elevation difference between the toe of the landslide and the lower riverside of the Minjiang River is 220 m (Fig. 2a).

The medium-height mountains and river valleys are developed in the landslide area. The majority of this area is a part of the Minshan Mountain of the Qionglai Mountains, with the southeast border belonging to the final segment of the Longmen Mountains (Wang et al., 2019). This area shows steep and dangerous valleys and slopes, narrow river valleys, and



deeply downcutting rivers. The Minjiang River flows through the area in a nearly N-E direction. The Tizicao landslide body
mainly consists of silty clay ($Q_4^{del}$) on the surface and broken phyllite below, while the sliding bed mainly comprises
carbonaceous phyllite of the Devonian Weiguan Group ($Dwg^2$). The carbonaceous phyllite is weak and broken. It has poor
physical and mechanical properties and poses risks of failure, slide, and deformation during the rainy seasons.
According to the field survey by Zhou et al. 2022, the middle-front part of the Tizicao landslide began to deform in
2013, when the houses on the slope started to crack and dislocate downward. In September 2014, the middle part of the
landslide front gradually collapsed. As a result, a flow area with a width of about 60 m and a height of 200 m (Fig. 2d) was
formed. The accumulation body fell into the Minjiang River to form a landslide dam. Meanwhile, the landslide rear (Fig. 2c)
began to crack. From August to September 2015, the landslide deformed more obviously and severely, resulting in additional
wider and longer cracks. After continuous deformation during the rainy season in 2016–2017, the rear and front of the
landslide had dislocated downward more than 10 m locally by July 2017. The collapse with a volume of about $6.0 \times 10^4$ m$^3$
occurred in the north front of the landslide body, blocking the Minjiang River for several hours. Fortunately, no casualties
occurred. Since October 2017, the deformation of the landslide has slowed down and tended to stabilize (Fig. 3c). However,
once the large-scale slide has occurred, this landslide would directly threaten the lives of 30 people on the slope body. It
would even seriously threaten the lives of 113 people in Shidaguan Town (Fig. 2a) below the landslide. More than 30
buildings and the National Highway G213 with 2 kilometers also would be destroyed.
The landslide can be divided into three areas according to the aerial photogrammetry of high-performance unmanned
aerial vehicles, field investigations, and monitored deformation data (Figs. 2b and 3a):
(a)  North sliding area. This area is rectangular and covers an area of $11.15 \times 10^4$ m$^2$. It has a longitudinal length of 600
m, and a transverse width of 258 m, and a sliding direction of 78°. A notable sliding failure has occurred in this area, as
shown in Fig. 4a. Specifically, significant tensional deformation has occurred at the landslide rear in this area, forming a rear
wall with a height of about 10 m (Fig. 2c). A deep and large tension crack (crack L04, Fig. 3a) and a pinnate shear crack
(crack L11, Fig. 3a) have developed in the middle part of this area. The overall sliding displacement is about 10.5 m. The
slope in the landslide front has suffered the most severe deformations. It has dislocated downward up to more than 40 m,
with about $6 \times 10^4$ m$^3$ of landslide masses having collapsed into the Minjiang River (Fig. 2d).
(b)  Middle deformation area. This area is in a long strip shape and covers an area of $6.42 \times 10^4$ m$^2$. It has a longitudinal
length of 568 m, a transverse width of 138 m, and a sliding direction of 78°. As shown in Fig. 4c, the landslide rear in this
area shows severe deformations, resulting in the formation of a 5 m high rear wall (crack L07, Fig. 3a). Multistage cracking
and depression deformations (cracks LF03, LF05, LF06, and LF07; Fig. 3a) have developed in the middle part of the
landslide in this area, with an overall displacement of about 8 m. Compression cracks and bulge-induced cracks (Fig. 3a)
have formed in the landslide front under the resistance of locking masses.
(c)  South deformation area. This area is in a long strip shape and covers an area of $13.21 \times 10^4$ m$^2$. It has a longitudinal
length of 700 m, a transverse width of 192 m, and a sliding direction of 85°. As shown in Fig. 4d, the landslide rear in this
area is controlled by cracks L07 and L08, which has dislocated downward about 3 m. The displacement of the middle part of
the landslide is about 1.5 m. The compression-induced longitudinal tension cracks (Cracks L09 and L10; Fig. 3a) have
mainly developed in the landslide front in this area, while large-scale sliding has not occurred.
Zhou et al. (2022) identified and analyzed the locking segment of the Tizicao landslide. According to the analytical
results, the locking segment lies at the south slope toe (Figs. 2a-b; Fig. 3b) based on the analysis of landform,
spatial-temporal deformations, surface cracks, and rock quality in the landslide area. It covers an area of about $4.69 \times 10^4$ m$^2$,
accounting for 15.2% of the total area of the landslide. As shown in Fig. 4, the anti-dip carbonaceous phyllites of the
Devonian Weiguan Group ($Dwg^2$) develop in the landslide area, which have different deformation characteristics in the
locking segment and the non-locking segment subjected to landslide deformation and unloading effect. For the locking
segment in the landslide area, the surface layer is a loose accumulation body, which is mainly composed of grayish yellow



silty soil mixed with fragments, with a thickness of about 3–5 m. The lower sliding body are carbonaceous phyllites (Fig. 5a),
which are moderately weathered to slightly with an attitude of 190°–260°∠36°–60°. The sliding bed is composed of the
slightly weathered carbonaceous phyllites with an attitude of 190°–260°∠60°–80°. The bedding planes of the slightly
weathered carbonaceous phyllites are straight and smooth, without filling or with a small amount of quartz veins, and are
hard structural planes. The joint spacing is 0.05–1.2 m, and the RQD value of rock mass is between 68.0% and 76.8%. The
anti-dip phyllites have a tendency of deformation along the slope direction, and the dip angles of the bedding plane decrease
gradually with the decrease of depth (Fig. 6). Furthermore, the anti-dip phyllites are less affected by landslide deformation
with the increase of the depth. For example, at 50m depth of the drilling borehole (Fig. 6c), the rock core integrity revealed is
good, and the rock mass strength is high, and the attitude is consistent with the slightly deformed anti-dip rock mass at the
back of the landslide (Fig. 5b). According to stereographic projections (Figs. 6d–f), with the increase of depth, the dip angle
of the bedding plane increases, and the stability of landslide increases. Correspondingly, the anti-dip rock mass is similar to
the rock bridge, which is called the locking segment herein, and is the key block to prevent further landslide sliding. Only
when the locking rock mass is cut off, dose the overall landslide failure occur.

14        For the non-locking segment in the landslide area, the surface layer is composed of the grayish yellow silty soil mixed

with fragmentary stone, the thickness of which is about 6–8 m. Below the surface layer there are the strongly weathered
carbonaceous phyllites with the thickness of 25–33 m. The sliding zone soils can be observed in the non-locking segment,
the thickness of which is about 0.5–1.2 m. There are the moderately weathered phyllites with an attitude of
252°–260°∠65°–73° below the sliding zone, and the joint space is 0.5–1.2 m, and the RQD value is 15.0%–54.5%. Due to
the large deformation in the non-locking segment, the deformation of the phyllites is also more severe, which is manifested
as the sliding of the phyllites along the slope direction after the toppling of the rock stratum (Figs. 7b–c,Fig. 8a). As shown
in Fig. 8, within the range of drilling depth, the bedding plane of rock mass (depth of 0–13 m) is slope direction, while the
rock mass below the depth of 13 m is counter-inclined. The attitude of rock mass below the sliding surface is basically
consistent with the original attitude. Therefore, the shearing failure of the anti-dip phyllites is the fundamental cause of the
large deformation in the north sliding area (Fig. 3a).
**3. Methods**
**3.1 Rock bridge models in the simulation program**
The FLAC3D program is used to simulate the 2D and 3D stability and deformation of landslides (Titti et al., 2020; Zhang et
al., 2013; Zhou et al., 2020). To investigate the 3D stability and deformation behaviors of the Tizicao landslide, three rock
bridge models were introduced into the FLAC-3D program. They are the intact rock mass model (IRMM; Kemeny, 2005;
Zhang et al., 2020), the Jennings model (JM; Bonilla-Sierra et al., 2015; Jennings, 1970), and the contact surface model with
high strength parameters (CSM-HSP; Huang et al., 2015; Scholtès and Donze, 2015), as shown in Fig. 9.

32        The IRMM model (Fig. 9a) is used to simulate the deformation and failure characteristics of rock bridges in rock

masses. This model can effectively reveal the behaviors of stress concentration, cracking, extension, and penetration (Tang et
al., 2001; Zhang et al., 2006). In the simulation of a landslide with a locking segment, an intact rock mass is used to simulate
a rock bridge (S1). The contact surface model is used to simulate the sliding surface (S2). The sliding body (Block A) and the
sliding bed (Block B) are linked with a continuous rock bridge (S1).

37        For the JM model (Jennings, 1970), the limit equilibrium method is initially employed to calculate the 2D stability of

rock slopes with discontinuous joints. In detail, the slope stability is calculated by assigning the equivalent shear strength
corresponding to different penetration rates to the potential sliding surface. The equivalent shear strength parameters can be
calculated as follows:



$c_{eq} = (1-k)c_r + kc_j$ (1)
$\tan \varphi_{eq} = (1-k)\tan \varphi_r + k \tan \varphi_j$ (2)
where, $c_{eq}$ and $\varphi_{eq}$ are the equivalent cohesion and the equivalent friction angle, respectively; $\varphi_r$ and $\varphi_j$ represent the friction
angle of intact rock and joints, respectively, and $c_r$ and $c_j$ are the cohesion of intact rock and joints, respectively.
Considering that co-planar joints are separated by the intact rock bridge, the relative quantity of intact rocks along the
sliding surface can be expressed by the ratio $k$, which is defined as (Jennings, 1970):
$k = \dfrac{\sum A_j}{\sum A_j + \sum A_r} = 1 - k_L$ (3)
where, $\sum A_j$ denotes the surface area of joints, $\sum A_r$ is the surface area of the rock bridge, and $k_L$ is the locking ratio (the ratio
of the surface area of the rock bridge to the total sliding surface area).
The Fos can be calculated using equation (4) below:
$Fos = \dfrac{\tau_f}{\tau} = \dfrac{N \tan \varphi_{eq} + c_{eq} A}{F_g \sin \theta}$ (4)
Where, $\tau_f$ is the shear force along the joint surface with normal force $N$, $A$ is the sliding surface area, $\theta$ is the inclination angle
of the planar surface, and $\tau$ is the sine component of the gravitational force $F_g$.
Some researchers (Bonilla-Sierra et al., 2015; Scholtès and Donze, 2015) introduced the Jennings model into the 3D
plane sliding analysis of the slope with rock bridges. They concluded that the rock bridges have notable control effects on the
stability and failure of the slopes. However, the stability of true 3D landslides with a locking segment is to be further studied.
Herein we introduced the Jennings model into the FLAC3D program. Then we simulated the 3D stability of the whole
landslide by assigning equivalent shear strength parameters to the contact surface model (S3), as shown in Fig. 9b.
As shown in Fig. 9c, two contact surface models with high and low strength parameters each were used to simulate the
rock bridge (S4) and sliding surface (S5), respectively. The strength parameters of an intact rock mass are adopted for the
rock bridge. In addition, shear stiffness and normal stiffness higher than those of the sliding surface (Huang et al., 2015) are
required in the CSM-HSP model to simulate the real resistance characteristics of the rock bridge.
**3.2 3D stability simulations**
The 3D mesh model of the Tizicao landslide (Fig. 10) was established using the FLAC3D program. It was composed of a
sliding bed, a sliding body, and a sliding surface, with a length of 1,100 m, a width of 700 m, and a height of 800 m. In this
model, the sliding bed and sliding body were established using tetrahedral elements. The sliding surface was established
using contact surface elements, which allows the contact surface to slide. The geometric size and shape of the 3D sliding
surface were deduced according to the depth of the sliding zone soil obtained by drilling. The parameters such as the area
and the position of the locking segment were obtained by Zhou et al. (2022). The constitutive model of Mohr-Coulomb was
used in the simulation. The bottom displacement was fixed as a boundary. The top surface was set as a free boundary. The
other four surfaces were set as boundaries with perpendicular fixed displacement. Given that the simulation in this study is
only aimed at exploring the deformation and the overall stability of the landslide, the sliding body and sliding bed were
supposed to be homogenized. The factors, such as joints and heterogeneity of rock masses, were temporarily not considered.
The simulation parameters of the sliding body, sliding bed, and sliding surface in the model were obtained through indoor
geotechnical tests. The simulation parameters are shown in Table 1.
The simulation analysis of the Tizicao landslide was conducted using the above three rock bridge models. As revealed
by site drilling, the rock masses in the locking segment have the same integrity as the phyllites in the sliding bed. Therefore,
the strength parameters of the rock bridges were set as those of the rock masses in the sliding bed (locking masses) in the
IRMM model. Meanwhile, the shear stiffness and normal stiffness of the sliding surface in this model were both set at 2.0



MPa/m to simulate the sliding state of the landslide. For the JM model, the rock bridge and sliding surface were both
simulated using the contact surface model. According to the site survey, the area of the locking segment accounts for 15.2%
of the total area of the landslide. The equivalent internal friction angle and equivalent cohesion were obtained to be 35.68°
and 503.24 kPa, respectively by solving equations (1) and (2) individually. The tensile strength, shear stiffness, and normal
stiffness of the sliding surface were set at 0.18 MPa, 1800 MPa/m, and 1800 MPa/m, respectively in the JM model. For the
CSM-HSP model, the locking masses were replaced with the contact surface model of which the strength and stiffness were
both higher than those of the sliding surface. Their strength parameters were set as those of the sliding bed. Meanwhile, the
shear stiffness and normal stiffness of the contact surface of the rock bridge were both set at 2000 MPa/m. The strength
parameters and stiffness coefficients of the sliding surface in the CSM-HSP were set the same as those of the sliding surface
in the IRMM.
**3.3 2D stability simulation**
To compare the differences between the 2D and 3D stability of the landslide, the 2D stability analysis of the Tizicao
landslide was performed on the four sections (Fig. 4) using the SLOPE/W module of the program GeoStudio 2012.
Meanwhile, the JM model was introduced into Bishop's algorithm of the GeoStudio program. The equivalent shear strength
parameters were determined based on penetration rates. Then they were assigned to the sliding surface to calculate the 2D
Fos. The simulation parameters of the sliding body, sliding surface, and locking mass are shown in Table 1. According to the
site survey, the locking ratios $k_L$ of sections A-A', B-B', C-C', and D-D' are 0, 0, 0.23, and 0.26, respectively, and the
calculated 2D stability factors are shown in Table 2 and Fig. 11.
**4. Results**
**4.1 Comparison analysis of 2D and 3D stability**
Table 2 shows the obtained 3D Fos of the Tizicao landslide using the three models and the 2D Fos of the landslide calculated
by using the JM model. The obtained 3D Fos using the IRMM, JM, and CSM-HSP models are 1.780 ± 0.2, 1.950 ± 0.3, and
1.710 ± 0.2, respectively. They are almost equal and their average value is 1.813. This indicates that the Tizicao landslide is
stable and large-scale sliding will not occur under the current conditions. The state of the landslide is consistent with the
displacement data monitored in the field (Fig. 3c). As shown in Fig. 11, the 2D Fos of sections A-A', B-B', C-C', and D-D'
of the Tizicao landslide are calculated to be 0.978 ± 0.15, 0.924 ± 0.1, 1.888 ± 0.23, and 2.075 ± 0.20, respectively.
Therefore, the landslide is unstable along sections A-A' and B-B', consistent with the large-scale collapse in the northern
front of the landslide (Fig. 2d). In contrast, the landslide is stable along sections C-C' and D-D', which agrees well with the
middle and south deformation area of the landslide. The difference in the landslide stability between the northern (sections
A-A' and B-B') and southern (sections C-C' and D-D') sides of the landslide is primarily caused by the existence of the
locking masses in the southern front of the landslide (Fig. 2b). According to Table 2, the 3D Fos of the Tizicao landslide are
greatly different from the 2D Fos. The latter can only represent the local stability rather than the overall stability of the
landslide.
**4.2 Analysis of landslide deformations**
According to the above analysis, all the IRMM, JM, and CSM-HSP models can be used to effectively simulate the overall
stability of the 3D landslides and obtain 3D stability factors. However, the JM model cannot simulate the real 3D
deformation behaviors since it is a model using equivalent strength parameters. Meanwhile, the IRMM model is rather
complex in modeling although it can be used to obtain real 3D deformation characteristics of landslides. Therefore, the
CSM-HSP model was selected to simulate the deformation trends of the Tizicao landslide. Figs. 12a−d show the total





displacement contours of the sliding body, the shear displacement contours and the sliding state of the sliding surface, also
the sliding velocity vectors of the sliding surface, respectively.
As shown in Fig. 12a, the total displacement contours of the sliding body showed notably different deformation zones.
They are the intense deformation zone at the rear and on the north side wall of the landslide, the moderate deformation zone
in the middle part of the landslide and the northern part of the landslide front, and the slight deformation zone in the middle
and southern parts of the landslide front. The maximum displacement is 10.69 m at the rear of the landslide (Fig. 12a), which
is agreed with the crack width of L07 (Fig. 13). As shown in Fig. 12a, owing to the sliding resistance effect of the locking
segment, the Tizicao landslide tends to slide northeastward in general. This tendency is consistent with the crack distribution
(Fig. 3a) and the isoline map of the surface displacement (Fig. 3b). As shown in Fig. 12a and Fig. 3a, the displacement exists
difference, just because the monitoring data was obtained from August 13, 2017, to January 25, 2018 after the large
deformation occurred (July 2017), which was not the whole deformation data for the landslide. However, the deformation
tendency in Fig. 12a is the same as in Fig. 3a.
Fig. 12b shows that the shear deformation of the sliding surface agrees well with the total displacement contours (Fig.
12a). It can be observed that the shear displacement of the sliding surface is 0 at the position of the locking segment. Fig. 12c
shows the sliding state when the Tizicao landslide is in equilibrium under the current conditions. The red, blue, and green
zones in Fig. 12c represents the sliding surface areas where sliding has not occurred, is occurring and has occurred,
respectively. Therefore, the locking segment on the sliding surface has no shear displacement. Then the 3D locking segment
along the sliding surface can be observed. The sliding velocity vector diagram of the sliding surface (Fig. 12d) indicates that
the sliding velocity is small and tends to be 0 in the locking segment. Therefore, the existence of the locking segment is the
fundamental reason why the landslide has not suffered large-scale sliding so far as a whole.
**5. Discussion**
**5.1 Effects of the locking ratios on 3D stability**
To establish the landslide models with different locking ratios, rectangular wireframes were used to cover the outline of the
landslide (Fig. 14), and the length and width of the wireframes and their ratios were obtained. Rectangles with increasing
length and width but fixed length/width ratio were used to gradually match the landslide from the southern part of the front
to the rear in the north. Then the coverage areas and positions of the 3D sliding surface were obtained with the actual locking
ratio changing from 0 to 1 (interval: 0.1). Afterwards, the 3D modeling of the Tizicao landslide was carried out using the
three rock bridge models according to the coverage areas and positions of the 3D sliding surface.
Fig. 15 shows the 3D Fos curves of the landslide under different locking ratios. It can be found that the 3D Fos curves
obtained using the three rock bridge models were roughly the same. In detail, they are parabolas overall and all the Fos first
increase and then tend to be stable as the locking ratio increases. The actual locking ratio of the Tizicao landslide is 0.152
according to the field survey, while the corresponding 3D stability factor is 1.71–1.95. In the case that the locking area of the
landslide gradually decreased to 0 (no locking segment), the 3D Fos of the landslide will be 1.215, which would decrease by
29.0 %–37.7 % compared to the 3D Fos under current conditions. In this case, the landslide tends to be unstable. This
indicates that the locking segment has significant effects on the overall stability of the landslide.
According to Equation (4), there is a linear relationship between the locking ratio and the stability factor, which applies
to the 2D stability of planar sliding landslides (Jennings, 1970). However, the Fos of 3D landslides with a locking segment
varied with the locking ratio in the form of an approximate quadratic parabola (Fig. 15). These variation trends are caused by
the effects of the positions of the locking masses and the curvature of the sliding surface.
As shown in Fig. 15, the 3D Fos curves are notably piecewise. There are two linear fitting curves (black dashed lines)
for the 3D Fos. The varying rate of 3D Fos under a locking ratio of less than 0.6 is significantly higher than that under a



locking ratio of greater than 0.6, which is about six times. Therefore, in the case of a high locking ratio of the landslide (a
small penetration rate), the changes of the locking ratio would have a small impact on the overall stability of the landslide. In
contrast, when the locking ratio of the landslide decreases to less than 0.6, the overall stability of the landslide significantly
decreases with a decrease in the locking ratio. This is the immediate cause of the fact that the Fos of the landslide rapidly
decreases and suffers a dramatic failure under a critical failure condition.
**5.2 Effects of the strength parameters of sliding surface and locking masses on 3D stability**
To estimate the effects of the strength parameters on 3D stability of the landslide, the strength parameters of sliding surface
and locking masses were obtained by the direct shear test for the drilled soil or rock cores. The cohesion and the internal
friction angle of the locking masses are 10–20000 kPa and 20–65°, respectively, and the cohesion and the internal friction
angle of the sliding surface are 6–1000 kPa and 5–35°, respectively. Then the 3D stability factor curves under different
strength parameters with a locking ratio of 0.5 were obtained using the three rock bridge models, as shown in Fig. 16. As
shown in Fig. 16a, the 3D Fos rapidly increases, and then shows a stable value, when the cohesion of locking masses is
10–1000 kPa, and greater than 1000 kPa, respectively. Therefore, the 3D Fos is sensitive to the cohesion of locking masses
with 10–1000 kPa but does not significantly vary when the cohesion is greater than 1000 kPa. The effects of the cohesion of
the sliding surface on the 3D Fos exhibited different characteristics (Fig. 16c). The 3D Fos obtained by using the IRMM and
CSM-HSP first increases non-linearly with the increase of the cohesion of sliding surface, and then shows a stable value
while the 3D Fos obtained by using the JM increased with the cohesion of sliding surface, but the acceleration rate decreases.

18        As shown in Fig. 16b, the 3D stability factor of the landslide non-linearly increases first and then tends to be stable with

an increase of the internal friction angle of locking masses. It increases from 2.49 to 4.53 (1.82 times) as the friction angle of
the locking mass increases from 20° to 65°, and the average growth rate is 0.045. The 3D Fos of the landslide varies with the
internal friction angle of the sliding surface in a similar trend (Fig. 16d). However, the 3D Fos increases from 3.20 to 4.58
(4.13 times) as the internal friction angle of the sliding surface increases from 5° to 35°, and the average growth rate is 0.046.
According to the comparison of the average growth rate, the internal friction angle of the locking masses and the sliding
surface has almost the same influence on the 3D Fos of the landslide.
**5.3 Comparative analysis of the three rock bridge models in the numerical simulation program**
The 3D stability factors obtained using the IRMM, JM, and CSM-HSP models were almost equal (Fig. 15). It is indicated
that all these three models can be used to effectively simulate the overall stability of a landslide with a locking segment. The
IRMM model (Fig. 9a) is frequently used to simulate the stability and the deformation and failure behaviors of 2D and 3D
rock slopes with rock bridges (Zhang et al., 2014; Hu et al., 2018). It can simulate the actual deformation process of the
slopes and is one of the most effective models used to simulate rock slopes/landslides. However, accurate information such
as the area and position of a locking segment are required in the IRMM model. Thus, it is necessary to identify a locking
segment of the landslides in detail before the stability analysis of landslides. Furthermore, it is quite difficult to identify the
locking segment of landslides due to the concealment of the locking masses (Elmo et al., 2018; Guerin et al., 2019).
Meanwhile, the uncertain positions and irregular geometric size of a locking segment also bring great difficulties to landslide
modeling. The JM model (Fig. 9b) cannot be used to further analyze the deformation and failure behaviors of landslides and
obtain actual deformation since it ignores the positions of rock bridges and the response of rock bridges to the landslide
deformation (Einstein et al., 1983). However, the 3D stability factor of landslides (Figs. 15−16) can be obtained using this
model. Therefore, it is reasonable to use the JM model to only analyze the macroscopic 3D stability of landslides. For the
CSM-HSP model (Fig. 9c), the contact surface models with high and low strength parameters each are used to simulate the
rock bridge and the sliding surface, respectively. It combines the advantages of the IRMM model in simulating the actual
deformation of slopes with rock bridges and the modeling advantages of the JM model. With this model, not only the overall





deformation and Fos of landslides can be obtained, but also the position and area of a locking segment can be changed at will,
thus greatly reducing the workload in the modeling of landslides with rock bridges. The CSM-HSP model performs better in
simulating both the 3D stability and the deformation and failure behaviors of landslides with a locking segment among the
three rock bridge models.

## 6. Conclusions

All the IRMM, JM, and CSM-HSP models can be used to obtain the 3D Fos of the landslide with a locking segment.
These models provide a convenient and effective simulation approach for assessing and predicting the 3D stability of the
landslide with a locking segment, respectively. The simulation results indicate that the Tizicao landslide is stable overall
under the current conditions, owing to the existence of the locking segment in the southern front, which are consistent with
the deformation and failure characteristics, the position and area of the locking segment, and the site monitoring data of the
landslide. Through the comparison results of 3D stability and 2D stability of the Tizicao landslide, we can conclude that the
2D stability is only suitable for the local stability analysis while the 3D stability represents the overall stability state of the
landslide with a locking segment. The discussion shows that there is a linear relationship between the locking ratio and the
stability factor of the landslides for the 2D stability of planar sliding landslides with a locking segment, while an
approximate quadratic parabola for the 3D stability of landslides with a locking segment under the effects of the positions of
the locking masses and the curvature of the sliding surface. The growth of the strength parameters of the locking segment
and the sliding surface can non-linearly increase the stability of landslides. The 3D Fos of the landslide is sensitive to the
cohesion of the locking segment and sliding surface of 10–1000 kPa. The internal friction angles of the locking masses and
the sliding surface have almost the same influence on the 3D Fos of the landslide. In the three rock bridge models, the
CSM-HSP combines the advantages of the IRMM model in simulating the actual deformation of slopes with rock bridges
and the modeling advantage of the JM model. It performs better in simulating both the 3D stability and the deformation and
failure behaviors of landslides with a locking segment among the three rock bridge models.
*Data availability.*    The landslide research data used in the paper are mainly derived from Zhou et al. (2022), as well as the site survey
conducted by our team.
*Competing interests.*    The authors declare that they have no conflict of interest.
*Author contribution.*    Yuntao Zhou developed the model code and performed the simulations, and wrote the manuscript draft; Xiaoyan
Zhao and Bernd Wünnemann reviewed and edited the manuscript; Guangze Zhang, Jiajia Zhang and Minghui Meng performed the
landslide investigations.
*Acknowledgments.*    This study was supported by the National Natural Science Foundation of China (grant no.: 41672295), Ministry of
Science and Technology of China (grant no.: 2019YFC1509904), and China Geological Survey (grant no.: DD20221745). The authors are
grateful to the editors and reviewers for kind and constructive suggestions.

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



Figure 2: Overall perspective of the site area of the Tizicao landslide (after Zhou et al. 2022). a An orthoimage of the landslide site area taken on November 24, 2020, with the pixel-sized of 3840 × 2160. b Three deformation areas of the Tizicao landslide. The red dashed line is the boundary of the deformation area. c Rear wall. d Rockslide area, flow area, and accumulation debris.



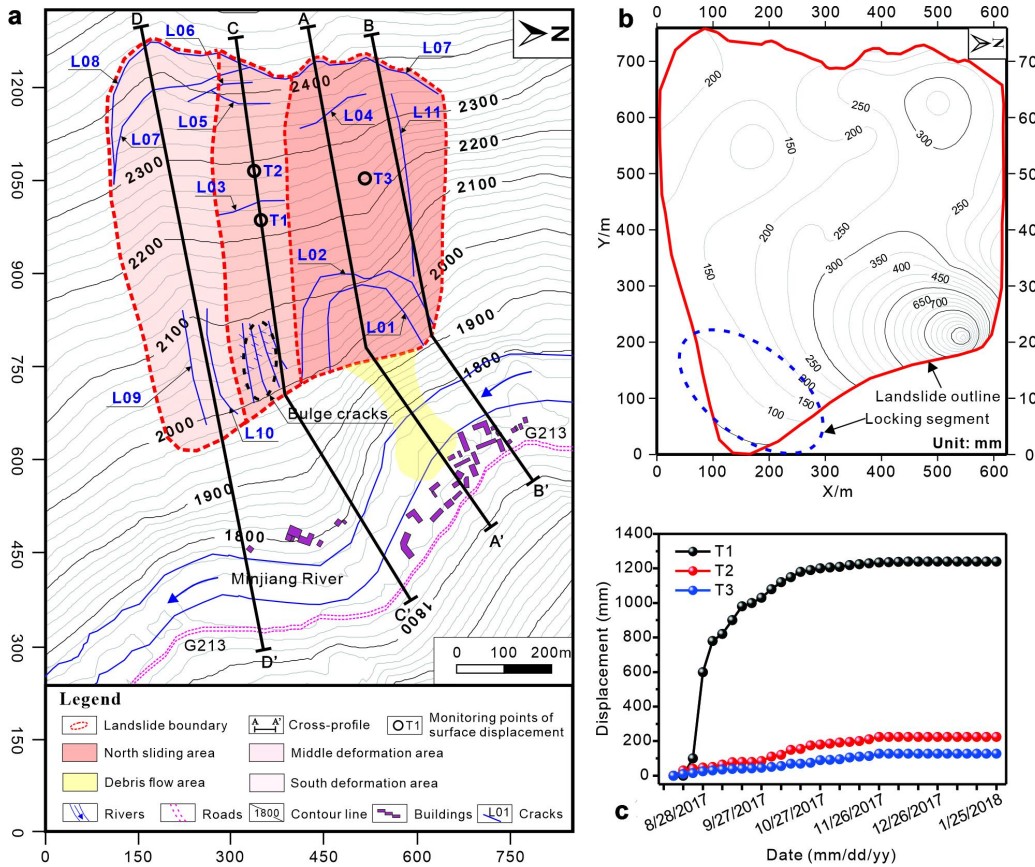

**Figure 3: Topographic plan, isoline map of the surface displacement, and the displacement monitoring curves of the Tizicao landslide. a Topographic plan of the deformation areas, the crack distribution, and the locations of engineering-geotechnical sections (after Zhou et al. 2022). b Isoline map of the surface displacement of the Tizicao landslide from August 13, 2017, to January 25, 2018 (Zhou et al. 2022); c Displacement monitoring curves of the landslide surface (8/13/2017–1/25/2018).**



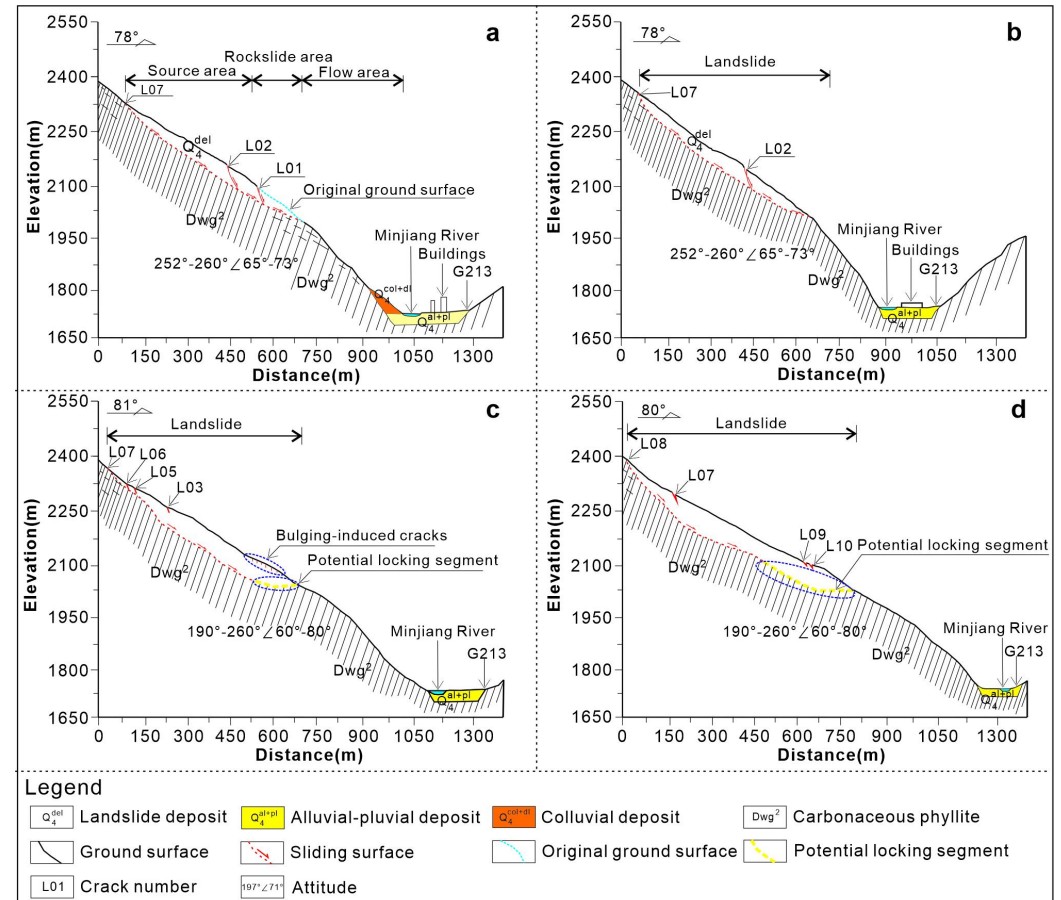

Figure 4: Engineering-geotechnical sections of the Tizicao landslide. a Section A-A'. b Section B-B'. c Section C-C'. d Section D-D' (after Zhou et al. 2022).

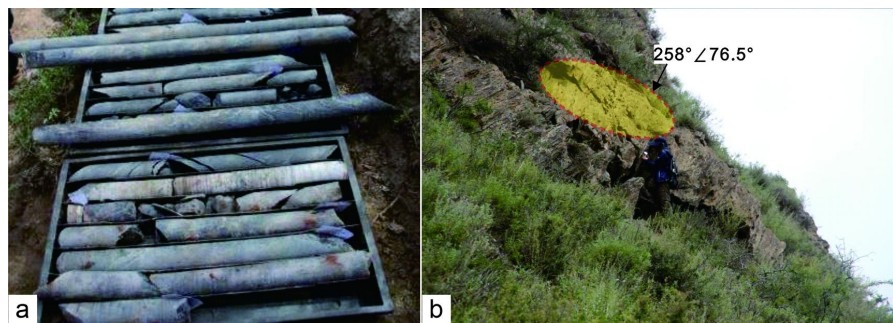

Figure 5: Rock cores drilled in borehole zk20 and exposed phyllites at the back of the landslide. a Intact rock cores (Zhou et al. 2022). b Exposed phyllites with an attitude of 258°∠76.5°.



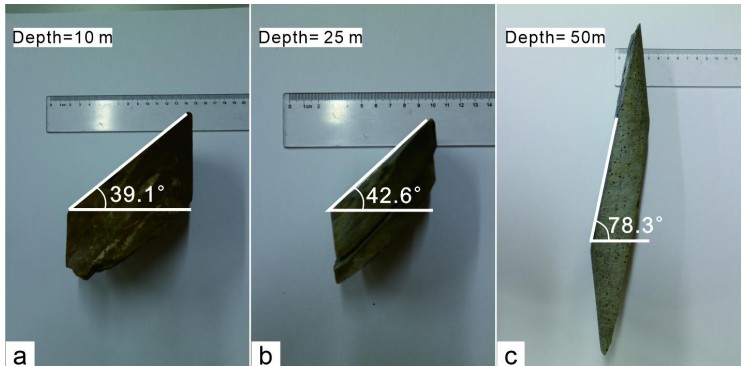

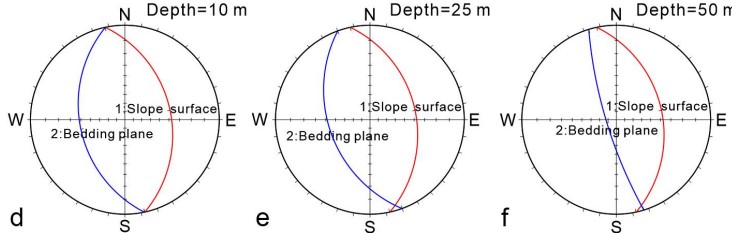

**Figure 6: Rock cores and stereographic projections at different depths. a–c Rock cores at the depth of 10 m, 25 m, and 50 m in borehole zk20, respectively. The diameters of the drilling hole are 100 mm and 60 mm at the depth of 0–15 m and 15–70 m, respectively. d–f Stereographic projections at the depths of 10 m, 25 m, and 50 m, respectively.**

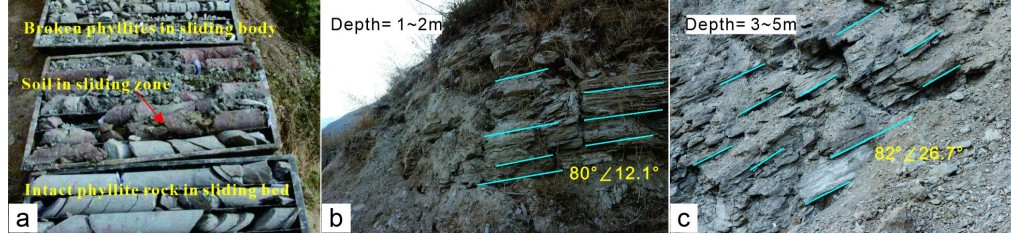

**Figure 7: Rock cores drilled in borehole zk08 and extremely broken phyllites exposed by construction excavation in the non-locking segment of the landslide. a Broken phyllites, soils in sliding zone, and relatively intact phyllites in sliding bed (Zhou et al. 2022). b Exposed phyllites with an attitude of 80°∠12.1° at the depth of 1–2 m. c Exposed phyllites with an attitude of 82°∠26.7° at the depth of 3–5 m. Cyan lines represent the bedding planes of the phyllites.**

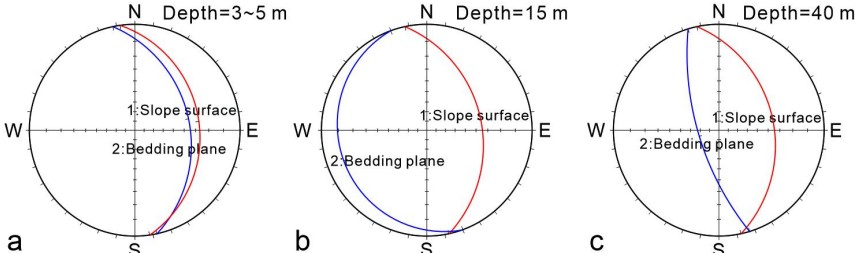

**Figure 8: Stereographic projections at the depths of 3-5 m, 15 m, and 40 m (a–c, respectively).**



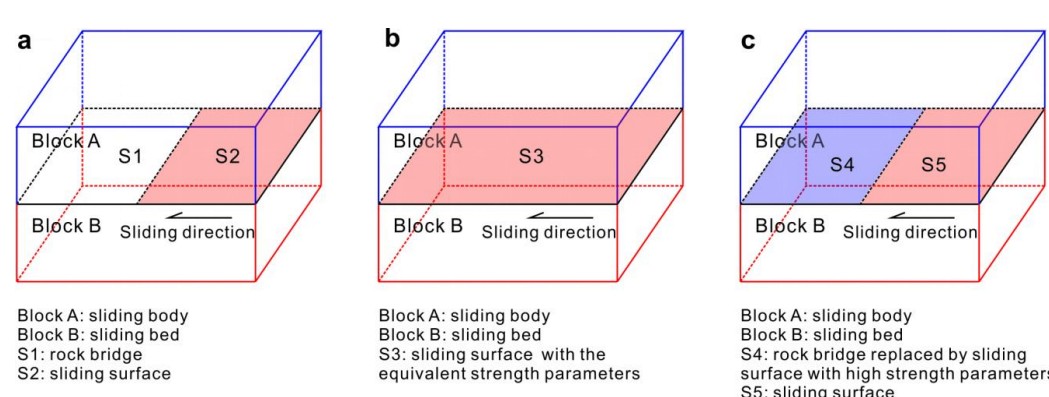

3  **Figure 9: Three rock bridge models used in the FLAC3D program. a IRMM. b JM. c CSM-HSP.**

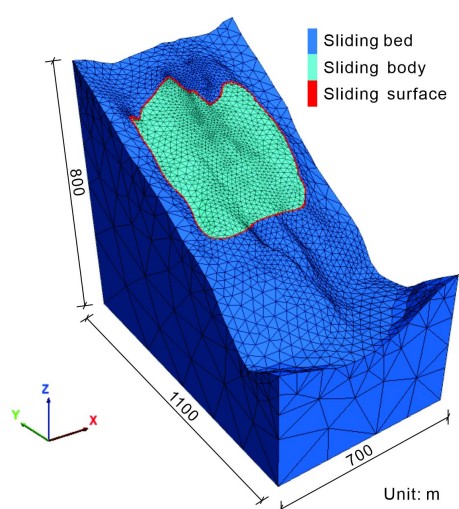

6  **Figure 10: The mesh model and geometry of the Tizicao landslide.**



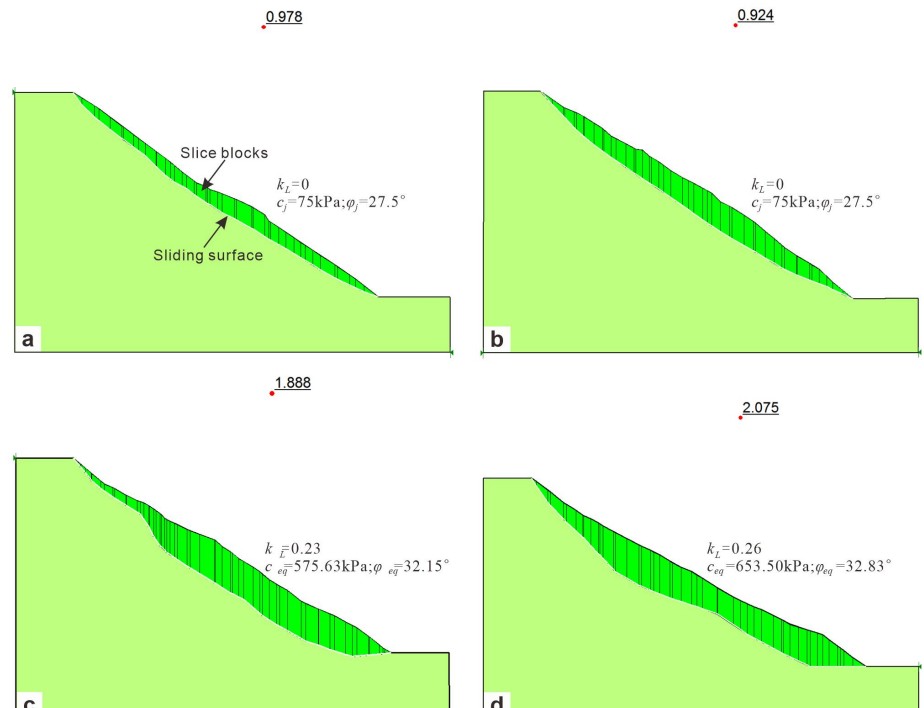

Figure 11: 2D Fos for different sections. a Section A-A'. b Section B-B'. c Section C-C'. d Section D-D'.



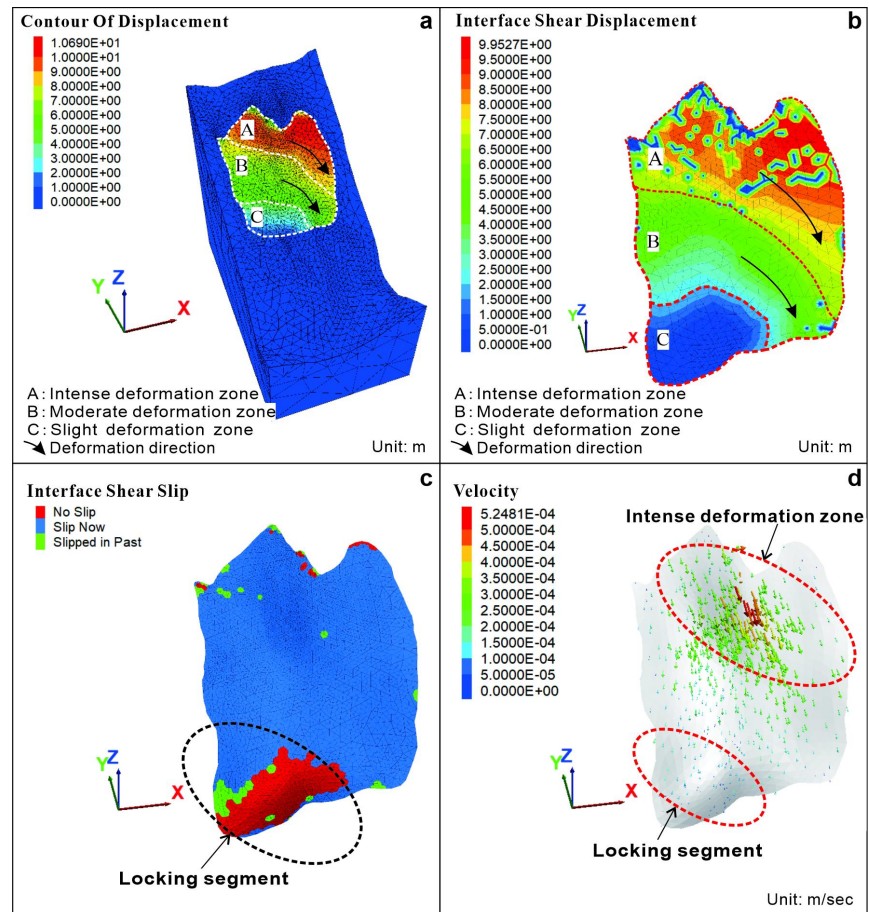

Figure 12: The simulation results of the Tizicao landslide. a Total displacement contours. b Shear displacement contours of the sliding surface. c Sliding state of the sliding surface. d Sliding velocity vectors of the sliding surface.

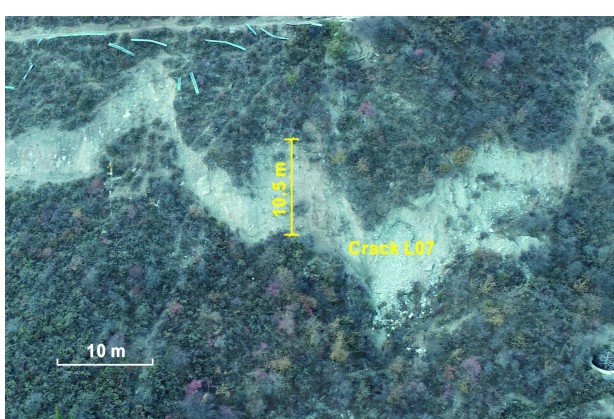

Figure 13: Crack L07 at the rear of the landslide. The crack width of L07 at the direction of the landslide is 10.5 m.



2    **Figure 14: Locking areas under different locking ratios.**

5    **Figure 15: 3D Fos curves under different locking ratios by using three rock bridge models.**

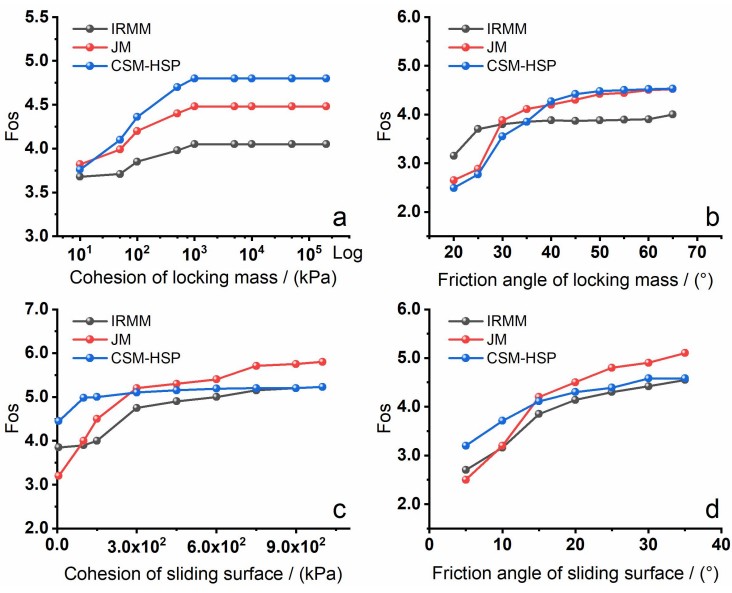

7    **Figure 16: 3D Fos curves under different strength parameters.**



1    **Table 1: Simulation parameters of the landslide model.**

| Model component | Density (g/cm³) | Young's modulus (GPa) | Poisson's ratio | Internal friction angle (°) | Cohesion (kPa) | Tensile strength (MPa) |
|---|---|---|---|---|---|---|
| Sliding body | 2.10 | 5.00 | 0.37 | 32.86 | 85.51 | 0.30 |
| Sliding bed | 2.72 | 40.00 | 0.30 | 37.00 | 580.00 | 1.04 |
| Sliding surface | - | - | - | 27.5 | 75.00 | 0.02 |

3    **Table 2: 2D and 3D Fos.**

| 2D/3D stability | Rock bridge simulation model | | Factor of safety (Fos) |
|---|---|---|---|
| | IRMM | | 1.780 ± 0.2 |
| 3D stability | JM | | 1.950 ± 0.3 |
| | CSM-HSP | | 1.710 ± 0.2 |
| | | Section A-A' | 0.978 ± 0.15 |
| | | Section B-B' | 0.924 ± 0.1 |
| 2D stability | JM | Section C-C' | 1.888 ± 0.23 |
| | | Section D-D' | 2.075 ± 0.20 |

