# Peer review of "Simulation analysis of 3D stability of a landslide with a locking 2 segment: A case study of the Tizicao landslide in Maoxian County, 3 Southwest China"

_EGUsphere, 2023_

## Author Comment (AC1)

**Comments on "Simulation analysis of 3D stability of a landslide with a locking segment: A case study of Tizicao landslide in Maoxian County, Southwest China"**

**RC1: 'Comment on egusphere-2023-28', Hendy Setiawan, 15 Apr 2023**

Your manuscript presents stability analysis of a landslide with a locking segment using three rock bridge models i.e., intact rock mass model, Jenning's model, and contact surface model with high strength parameters in the FLAC3D program. Presenting results discussing the effects of locking ratios, strength parameters of sliding surface and locking mass, as well as a comparison of three rock bridge models in addressing the 3D stability of this landslide. No significant objections came to this manuscript. However, some necessary amendments are needed:

**1. In conclusion, the simulation results indicate that the landslide is stable overall in current conditions, due to the existence of the locking segment, and is consistent with field deformation and monitoring data. But here we did not find any descriptions of what monitoring data and instruments were installed in the field. Please described.**

**Response 1:** Thank you for your so careful reading. In fact, we monitored the Tizicao landslide for several years. In the landslide body, twenty-four fixed non-prism monitoring points (T1−T24) were deployed to primarily monitor the surface displacement from June 1, 2017, to October 2, 2017, as shown in Fig. 8 in reference Zhou et al., 2022. They covered almost the entire landslide body. These raw data on the surface displacement were processed using the measurement adjustment software DDM to obtain their deformation amplitude and rates. The detailed descriptions of the monitoring data and instruments were presented in reference Zhou et al., 2022, as shown in the following figure:

[Figure]

**Fig. 1** Plan showing crack development and monitoring point layout in the Tizicao landslide. L01−L11 refer to the numbers of 11 cracks developing in the landslide; besides, there are great numbers of bulging-induced fissures in the middle part of the landslide front; T1−T24 represent 24 fixed non-prism monitoring points; D01−D07 represent seven boreholes used to monitor the deep displacement of the landslide (from Zhou et al. 2022)

The reference is listed as follows:

Zhou, Y. T., Zhao, X. Y., Zhang, J. J., Meng, M. H.: Identification of a locking segment in a high-locality landslide in Shidaguan, Southwest China, Nat. Hazards, 111, 2909–2931, https://doi.org/10.1007/s11069-021-05162-1, 2022.

**2. According to Figure 3, there are T1, T2, and T3 for the surface displacement monitoring points, but only T1 have been installed nearby the crack. Then why the contour or isoline map of surface displacement is concentrated on the northeast part of**

the landslide where there are no monitoring points (Figure 3b)? Please clarify and give further detailed and concise descriptions of these matters.

**Response 2:** Thank you for your so careful reading. In the manuscript, we did not describe all the monitoring points in Fig. 3a or the detailed monitoring data in Fig. 3c. As described in "Response 1", twenty-four fixed non-prism monitoring points (T1−T24) were deployed to monitor the surface displacement of the landslide, which was shown in Fig. 8 in reference Zhou et al., 2022. Then, the 24 monitoring curves were obtained to plot the isoline map of surface displacement (Fig. 3b). Figure 3b was also cited from reference Zhou et al., 2022. In Figures 3a and 3c, three monitoring points (T1, T2, and T3) were just used to show the deformation tendency of the landslide, also they are plotted in the isoline map of surface displacement.

**3. Methods section only presents numerical 2D and 3D simulations but not yet describing monitoring instruments and laboratory tests (if you use parameters from the tests for the numerical simulation). Please confirm.**

**Response 3:** Thanks for your suggestion. The monitoring instruments are not described in the manuscript because the monitoring instruments and the monitoring data are described in reference Zhou et al., 2022, which was cited in this study. We took the rock samples from the sliding body, sliding bed, and sliding surface of the landslide to conduct the geotechnical tests. The obtained rock parameters (Table 1) were used for the numerical simulation. Among them, the rock density was obtained using the wax-sealing method; the Young's modulus, Poisson's ratio, internal friction angle, and cohesion of rocks were collected from the triaxial test; and the tensile strength was obtained from the Brazilian test. We have added the descriptions of the laboratory tests in section "3.2 3D stability simulations" in the revised manuscript. Please see the text marked in red for details. Thanks.

**4. Page 5 Lines 34-35, "The simulation parameters of the sliding body, sliding bed, and sliding surface in the model were obtained through indoor geotechnical tests, shown in Table 1". What kind of geotechnical tests? Please describe.**

**Response 4:** Thanks for your suggestion. The obtained rock parameters (Table 1) were used for the numerical simulation. Among them, the rock density was obtained using the

wax-sealing method; the Young's modulus, Poisson's ratio, internal friction angle, and cohesion of rocks were collected from the triaxial test; and the tensile strength was obtained from the Brazilian test. We have added the descriptions of the laboratory tests in section "3.2 3D stability simulations" in the revised manuscript. Thank you.

**5. Page 6 Lines 4-5, the parameters of tensile strength, shear stiffness, and normal stiffness of the sliding surface for the JM model are assumed or based on the laboratory experiment. Please describe what kind of experiment.**

**Response 5:** For the JM model, the landslide was simulated by assigning equivalent shear strength parameters to the contact surface model (S3), as shown in Fig. 9b. Then, the rock bridge and sliding surface were assumed as homogeneous contact surfaces, so the parameters of the assumed contact surfaces were not set as the same parameters of the rock bridge or sliding surface. Then, we took the average values of the tensile strength of the rock bridge and the sliding surface, and the real tensile strength of the rock bridge and sliding surface were obtained from the Brazilian test. For the shear stiffness and normal stiffness of the sliding surface in the JM model, there was no direct reference yet, so we referenced the "FLAC3D6.0 Theory and Background". A good rule-of-thumb is that and be set to ten times the equivalent stiffness of the stiffest neighboring zone. The apparent stiffness of a zone in the normal direction is:

$$\max[\frac{K + \frac{4}{3}G}{\Delta z_{min}}]$$

where $K$ & $G$ are the bulk and shear moduli, respectively.

The [ ] notation indicates that the maximum value over all zones adjacent to the interface is to be used.

**6. Page 6 Lines 8-10, the strength parameters and stiffness coefficients of the sliding surface in the CSM-HSP and the IRMM were set the same. Could you please explain why?**

**Response 6:** For the IRMM, the sliding surface (S2) was replaced with the contact surface (Fig. 9a); for the CSM-HSP, the sliding surface (S5) was simulated using the contact surface

(Fig. 9c), too. Then, the sliding surface model and its simulation parameters were the same in the FLAC3D program. For the CSM-HSP and the IRMM, only the rock bridge model was different.

**7. Please consider the bar scale in the map of Sichuan Province in Figure 1.**

**Response 7:** Thank you for your suggestion. We have added the bar scale in Figure 1.

**8. Correct me if Figure 2b is an aerial/oblique view of the landslide? is it necessary to put a bar scale? I prefer to provide a bar scale for Figure 2a as it is an orthogonal image. Please confirm.**

**Response 8:** Figure 2b is an oblique view of the landslide. It is necessary to put a bar scale and we added it in Figure 2b. We also added a bar scale in Figure 2a.

**9. In Figure 4a, section A-A' the L07 should be L04, while in Figure 4b L11 not indicated in section B-B'. Then Figure 4c the monitoring points of T1 and T2 nearby L03 are not present. Please revised.**

**Response 9:** Thank you for your so careful reading. In fact, the L07 is not the L04 in section A-A' in Figure 4a, and we carelessly forgot to mark the L04, which has been added in Figure 4a. The L11 was not indicated in section B-B', and we have revised Figure 4b. In Figure 4c, we added the monitoring points of T1 and T2 nearby L03. Please see the revised Figure 4 for detail. Thanks.

**10. Location of borehole zk20 provided in Figure 5 is not shown on the map in Figure 3. Please add and clarify.**

**Response 10:** Thank you for your so careful reading. We have added the location of borehole zk20 in Figure 3 in the revised manuscript.

**11. Locations of zk08 in Figure 7a and exposed phyllites in Figures 7b and 7c are also not pointed on the map. Please add and clarify.**

**Response 11:** Thank you for your so careful reading. We added the location of borehole zk08 in Figure 3a. The exposed phyllites in Figures 7b and 7c have also been pointed in Figure 3a.

**12. Please provide a full description of each abbreviation in the caption of Figure 9.**

**Response 12:** Thank you for your suggestion. The full descriptions of abbreviations IRMM, JM, and CSM-HSP are the intact rock mass model, the Jennings model, and the contact surface model with high strength parameters, respectively. We have added the full description of each abbreviation in the caption of Figure 9.

**13. It is not clear whether the red zone of the sliding surface in Figure 10 is shown in the mesh model. Please clarify.**

**Response 13:** The red zone represents the sliding surface in the mesh mode in Figure 10. Because of the opacity of the sliding body mesh, the full view of the red zone of the sliding surface can't be observed. But we can see the full view in Figures 12b-d. Thanks.

**14. Cohesion between A-A', B-B' and C-C', D-D' in Figure 11 for 2D simulation are very different significantly, why? Check also its relation with simulation parameters in Table 1. Please discuss.**

**Response 14:** The differences in cohesion between A-A', B-B' and C-C', D-D' in Figure 11 for 2D simulation are caused by the locking ratio $k_L$ (the ratio of the surface area of the rock bridge to the total sliding surface area). For sections A-A' and B-B', the locking ratio $k_L = 0$, i.e., there was no rock bridge in these sections. Then, the shear strength parameters of the sliding surface can be set at the same values as those in Table 1. However, for sections C-C' and D-D', the locking ratios $k_L = 0.23$ and 0.26, respectively. According to the JM model, the slope stability wass calculated by assigning the equivalent shear strength corresponding to different penetration rates to the potential sliding surface. The equivalent shear strength parameters can be calculated as follows:

$$c_{eq} = (1-k)c_r + kc_j \tag{1}$$

$$\tan \varphi_{eq} = (1-k)\tan \varphi_r + k \tan \varphi_j \tag{2}$$

where $c_{eq}$ and $\varphi_{eq}$ are the equivalent cohesion and the equivalent friction angle, respectively; $\varphi_r$ and $\varphi_j$ represent the friction angles of an intact rock and joints, respectively, and $c_r$ and $c_j$ are the cohesion of an intact rock and joints, respectively.

Considering that co-planar joints are separated by the intact rock bridge, the relative quantity of intact rocks along the sliding surface can be expressed as the ratio $k$, which is defined as follows (Jennings, 1970):

$$k = \frac{\sum A_j}{\sum A_j + \sum A_r} = 1 - k_L \qquad (3)$$

where $\sum A_j$ denotes the surface area of joints, $\sum A_r$ is the surface area of the rock bridge, and $k_L$ is the locking ratio (the ratio of the surface area of the rock bridge to the total sliding surface area).

So, we used equivalent cohesion $c_{eq}$ and equivalent friction angle $\varphi_{eq}$ when calculating the 2D Fos. The equivalent shear strength parameters were obtained from equations (1) and (2) based on the shear strength parameters of the rock bridge and the sliding surface in Table 1, as well as the different locking ratios in Figures 11c-d. The equivalent shear strength parameters are shown in Figures 11c-d.

**15. Contours of shear displacement and intense deformation zone in Figure 12 seems not in line with the isoline map of surface displacement in Figure 3b. Perhaps unclear explanation in the text that I could not catch. Could you emphasize the above results?**

**Response 15:** Sorry for the unclear explanation of the deformation characteristics between the simulation results in Fig. 12 and the isoline map of surface displacement in Fig. 3b. We added the characteristics of the surface displacement in Fig. 3b, which are described as follows:

As shown in the isoline map of surface displacement (Fig. 3b), a sliding event occurred in a general northeast direction (closer to the north) from August 13, 2017 to January 25, 2018. In this event, the maximum surface displacement (1210 mm) occurred at the northern toe, which coincided with the location where the front collapsed (Fig. 2d). The landslide's rear and middle parts showed similar surface displacement of 150–300 mm in the sliding event, indicating that they slid as a whole. The minimum surface displacement of 30–150 mm occurred in the southern area of the slope toe throughout the whole sliding event. Therefore, the southern area serves as the anti-sliding area of the whole landslide.

Figs. 12a and 3a show different displacement values because the monitoring data obtained from August 13, 2017 to January 25, 2018 (after the large deformation in July 2017) do not include the complete deformation data of the landslide. However, Figs. 12a and 3a reflect the same deformation tendency.

**16. Font remark in Figure 13 is not visible, please change its color.**

**Response 16:** Thanks for your suggestion. We have changed the yellow color into black in Figure 13.

**17. The locking segment needs to be described further, is it generated due to the movement characteristic of a landslide? Or originally due to the lithological or geological conditions? Or are there any factors or settings that cause this feature? Please discuss.**

**Response 17:** Thanks for your suggestion. In section "2. Study site", we described the location, area, lithology, RQD values at different depths, and surface deformation characteristics of the locking segment. However, we did not describe the origin cause of the locking segment.

In fact, Zhou et al. (2022) preliminarily discussed the cause. The locking masses of the Tizicao landslide occur on the convex bank, while the non-locking masses have developed on the concave bank, indicating that the locking masses are directly related to the S-shaped river valley under the landslide. From a geomorphological point of view, landslides rarely occur on convex banks but occur more frequently on concave banks. From a topographical perspective, a convex slope is more stable than a concave slope under the same conditions. Noticeably, the concave and convex banks of the S-shaped valley under the Tizicao landslide differ greatly in slope and lithology. Therefore, the rock masses on the south side of the landslide above the convex bank are intact and constitute the potential locking segment of the landslide.

We added the above descriptions in section "2. Study site" to explain the cause of the locking segment.

---

## Author Comment (AC2)

**Comments on "Simulation analysis of 3D stability of a landslide with a locking segment: A case study of Tizicao landslide in Maoxian County, Southwest China"**

**RC2: 'Comment on egusphere-2023-28', Anonymous Referee #2, 16 Apr 2023**

Firstly, I would like to thank the Editor for giving me the opportunity to comment on this scientific article.

The paper focuses on the simulation analysis of the 3D stability of a landslide with a locking elements. The Tizicao landslide in Maoxian County, Southwest China represents a case study for the research. The locking elements plays a crucial role in the stability of the Tizicao landslide, acting as a barrier to sliding, thereby increasing the slope's stability. However, the locking elements can create stress concentrations and increase the risk of failure in other parts of the slope, emphasizing the importance of considering the presence and characteristics of locking elements when analyzing landslides. The authors attempt to numerically model a rock mass discontinuity problem through a continuum method.

However, there are several critical issues reported below:

**English**

**1. Firstly, the English language requires revision as it includes numerous errors and repetitions, and the form requires adjustment as it can be unclear and confusing at times.**

**Response 1:** Thank you for your valuable and thoughtful comments. We have asked a professional proofreading company to improve the writing of this manuscript, and a native English speaker has carefully checked and polished the language of this manuscript.

**Geomechanical characterization**

**2. Another issue is the geomechanical characterization. How was it conducted, and what tests were performed to acquire the mechanical parameters? For instance, the sentence "the equivalent shear strength parameters were determined based on penetration rates" on page 6, line 15, needs explanation about the method used to**

**determine these parameters.**

**Response 2:** Thank you for your suggestions. As for the geomechanical characterization, we collected rock samples from the sliding body, sliding bed, and sliding surface of the landslide to conduct the geotechnical tests. The rock density was obtained using the wax-sealing method; the Young's modulus, Poisson's ratio, internal friction angle, and cohesion of rocks were collected from the triaxial test; and the tensile strength was obtained from the Brazilian test. The obtained rock parameters (Table 1) were used for the numerical simulation. We have added the descriptions of the laboratory tests in section "3.2 3D stability simulations" in the revised manuscript. Please see the text marked in red for details. Thanks.

For the sentence "the equivalent shear strength parameters were determined based on penetration rates" on page 6, line 15, according to the JM model, the slope stability was calculated by assigning the equivalent shear strength corresponding to different penetration rates to the potential sliding surface. The equivalent shear strength parameters can be calculated as follows:

$$c_{eq} = (1-k)c_r + kc_j \tag{1}$$

$$\tan\varphi_{eq} = (1-k)\tan\varphi_r + k\tan\varphi_j \tag{2}$$

where $c_{eq}$ and $\varphi_{eq}$ are the equivalent cohesion and the equivalent friction angle, respectively; $\varphi_r$ and $\varphi_j$ represent the friction angles of an intact rock and joints, respectively, and $c_r$ and $c_j$ are the cohesion of an intact rock and joints, respectively.

Considering that co-planar joints are separated by the intact rock bridge, the relative quantity of intact rocks along the sliding surface can be expressed as the ratio $k$, which is defined as follows (Jennings, 1970):

$$k = \frac{\sum A_j}{\sum A_j + \sum A_r} = 1 - k_L \tag{3}$$

where $\sum A_j$ denotes the surface area of joints, $\sum A_r$ is the surface area of the rock bridge, and $k_L$ is the locking ratio (the ratio of the surface area of the rock bridge to the total sliding surface area).

So, the equivalent shear strength parameters were obtained from equations (1) and (2) based on the shear strength parameters of the rock bridge and the sliding surface in Table 1, as well as the different locking ratios in Figures 11c-d. The equivalent shear strength parameters are shown in Figures 11c-d.

**Numerical Modeling**

**3. The authors don't explain how the numerical analyses were set up in Flac3D. The**

**paper should clarify how the IRMM, JM, and CSP-HSP constitutive models were implemented in the Flac3D code. The reviewer is wondering if Perfectly Plastic Mohr-Coulomb model was modified.**

**Response 3:** We are sorry for the lack of clarification on the implementation of these models in the FLAC3D code. The IRMM, JM, and CSP-HSP models are simplified models of landslides with rock bridges. We replaced the rock bridges and the sliding surface with different elements (tetrahedral elements and contact surface elements) in the FLAC3D code. The details are described as follows:

As shown in Fig. 9a, in the simulation of a landslide with a locking segment, the rock bridge (S1), which is an intact rock mass, was simulated using the tetrahedral elements in the FLAC3D program, the sliding surface (S2) was simulated using the contact surface model in FLAC3D program, and the sliding body (Block A) and the sliding bed (Block B) were linked with the continuous rock bridge (S1).

[Figure]

**Figure 9: Three rock bridge models used in the FLAC3D program. a Intact rock mass model (IRMM). b Jennings model (JM). c contact surface model with high strength parameters (CSM-HSP).**

For the JM model, the limit equilibrium method is initially employed to calculate the 2D stability of rock slopes with discontinuous joints. Specifically, the slope stability is calculated by assigning the equivalent shear strength corresponding to different penetration rates to the potential sliding surface. In this study, we introduced the Jennings model into the FLAC3D program. Then, we simulated the 3D stability of the whole landslide (including the rock bridge and the sliding surface) by assigning equivalent shear strength parameters to the contact surface model (S3), as shown in Fig. 9b.

As shown in Fig. 9c, two contact surface models, one with high strength parameters and the other with low strength parameters, were used to simulate the rock bridge (S4) and sliding surface (S5), respectively. The strength parameters of an intact rock mass were adopted for the rock bridge. In addition, the shear stiffness and normal stiffness higher than

those of the sliding surface are required in the CSM-HSP model to simulate the real resistance characteristics of the rock bridge.

According to the above descriptions, we did not establish new constitutive models implemented in the Flac3D code. We just simplified the landslide models and provided different methods to simulate the landslide with rock bridges in the FLAC3D program using different mesh elements. In the simulation of the landslide using the IRMM, JM, and CSP-HSP models, all the mesh elements yielded a Plastic Mohr-Coulomb model, so the Perfectly Plastic Mohr-Coulomb model was not modified in this manuscript.

**4. The authors mention Geostudio as a software utilized for 2D calculations, however this sentence doesn't explain the 2D analysis method because Geostudio is a software suite which includes different products so the review is wondering if the numerical simulations were carried out with Limit Equilibrium Method (LEM) of the Slope/W software. The paper should explain how the JM model was introduced into the Bishop algorithm in LEM modelling.**

**Response 4:** Thank you for your comments. We conducted the 2D stability analysis of the Tizicao landslide using the SLOPE/W module of the program GeoStudio 2012. In the SLOPE/W module, there are many limit equilibrium methods for calculating the 2D Fos, among which we selected Bishop's algorithm for 2D calculations. Meanwhile, the JM model was introduced into Bishop's algorithm in the GeoStudio program. In the JM model, we only input the equivalent shear strength determined using equations (1) and (2) based on penetration rates for the sliding surface. In this way, the slope stability can be calculated. For sections A-A', B-B', C-C', and D-D', their locking ratios $k_L$ are 0, 0, 0.23, and 0.26, respectively according to the site survey. The equivalent shear strength was calculated, and the calculated 2D stability factors are shown in Table 2 and Fig. 11.

[Figure]

**Figure 2: 2D Fos for different sections. a Section A-A'. b Section B-B'. c Section C-C'. d Section D-D' (from the present study).**

**5. The paper should provide better specification of the comparisons between the four 2D surfaces and the hypothesized 3D surface.**

**Response 5:** Thank you for your suggestions. For the 2D sliding surfaces, they were deduced according to the depth of the sliding zone soil obtained by drilling, as shown in Fig. 4. Meanwhile, the geometric size and shape of the 3D sliding surface were deduced based on the four 2D sliding surfaces and the outline of the landslide. So, the 3D sliding surface was deduced from the four 2D sliding surfaces.

As for the comparisons between the 2D and 3D stability of the landslide, we added detailed comparisons between the four 2D stability and the 3D stability, as described as follows:

For the landslide sections with severe deformation (sections A-A' and B-B'), their 2D Fos values were lower than their 3D Fos values. However, for the landslide sections with slight deformation (sections C-C' and D-D'), their 2D Fos values were significantly greater than their 3D Fos values, especially for the landslide sections with the locking segment. The

relatively conservative 2D stability analysis (Li et al., 2010; Park et al., 2017) made the 2D Fos values usually lower than the 3D Fos values. Nonetheless, for the landslide sections with rock bridges, their 2D Fos values may exceed their 3D Fos values (Table 2). The overall stability of a landslide with rock bridges should be assessed using 3D Fos since the 2D Fos represents only the local stability of the landslide.

**Monitoring**

**6. Lastly, while the paper often discusses monitoring of deformations, it fails to explain how the monitoring was conducted and the tools used to obtain the data.**

**Response 6:** Thank you for your so careful reading. In fact, we monitored the Tizicao landslide for several years. In the landslide body, twenty-four fixed non-prism monitoring points (T1−T24) were deployed to primarily monitor the surface displacement from June 1, 2017, to October 2, 2017, as shown in Fig. 8 in reference Zhou et al., 2022. They covered almost the entire landslide body. These raw data on the surface displacement were processed using the measurement adjustment software DDM to obtain their deformation amplitude and rates. The detailed descriptions of the monitoring data and instruments were presented in reference Zhou et al., 2022.

---

## Author Response (AR2)

**Comments response on "Simulation analysis of 3D stability of a landslide with a locking segment: A case study of Tizicao landslide in Maoxian County, Southwest China"**

**RC1: 'Comment on egusphere-2023-28', Hendy Setiawan, 15 Apr 2023**

**1. In conclusion, the simulation results indicate that the landslide is stable overall in current conditions, due to the existence of the locking segment, and is consistent with field deformation and monitoring data. But here we did not find any descriptions of what monitoring data and instruments were installed in the field. Please described.**

**Response 1:** Thank you for your so careful reading. In fact, we conducted the monitoring work of the landslide for several years. The monitoring work was conducted by the Leica Monitoring Total Station (TM50), as shown in Fig. 1a below, and it was set up on the slope opposite the landslide. In the landslide body, twenty-four fixed non-prism monitoring points (T1−T24, Fig. 1b below) were deployed to mainly monitor the surface displacement from June 1, 2017, to October 2, 2017, as shown in Fig. 2 below in reference Zhou et al. (2022). They almost covered the entire landslide body. These raw data of the surface displacement were processed using the measurement adjustment software DDM to obtain their deformation amount and deformation rate. Furthermore, we added the descriptions of the monitoring work in "2. Study site". Please see the text marked in red in revised manuscript. Thanks.

[Figure]

**Fig. 1** Monitoring instruments. a. Leica Monitoring Total Station (TM50); b. Fixed non-prism monitoring points

[Figure]

**Fig. 2** Plan showing crack development and monitoring point layout in the Tizicao landslide. L01−L11 refer to the numbers of 11 cracks developing in the landslide; besides, there are great numbers of bulging-induced fissures in the middle part of the landslide front; T1−T24 represent 24 fixed non-prism monitoring points; D01−D07 represent seven boreholes used to monitor the deep displacement of the landslide (Zhou et al. 2022)

The reference is listed as follows,

Zhou, Y. T., Zhao, X. Y., Zhang, J. J., Meng, M. H.: Identification of a locking segment in a high-locality landslide in Shidaguan, Southwest China, Nat. Hazards, 111, 2909–2931, https://doi.org/10.1007/s11069-021-05162-1, 2022.

**2. According to Figure 3, there are T1, T2 and T3 for the surface displacement monitoring points, but only T1 have been installed nearby the crack. Then why the contour or isoline map of surface displacement is concentrated on the northeast part of the landslide where there are no monitoring points (Figure 3b)? Please clarify and give further detailed and concise descriptions of these matters.**

**Response 2:** Thank you for your so careful reading. In the manuscript, we did not describe all the monitoring points in Fig. 3a or the detailed monitoring data in Fig. 3c. As described in "Response 1", twenty-four fixed non-prism monitoring points (T1−T24) were deployed to monitor the surface displacement of the landslide, which was shown in Fig. 2 above in reference Zhou et al. (2022). Then the 24 monitoring curves were obtained to plot the isoline map of surface displacement (Fig. 3b), and the Figure 3b was also quoted from reference Zhou et al. (2022). According to the isoline map of surface displacement, the contour of surface displacement is concentrated on the northeast part of the landslide. In Figure 3a and Figure 3c, three monitoring points (T1, T2 and T3) were just used to show the deformation tendency of the landslide, not to plot the isoline map of surface displacement. To describe the monitoring work better in this landslide, we added the location of 24 fixed non-prism monitoring points (T1−T24) in Fig. 2b below.

[Figure]

**Figure 2: Overall perspective of the site area of the Tizicao landslide (after Zhou et al., 2022). a An orthoimage of the landslide site area taken on November 24, 2020, with a resolution of 3840 ╳ 2160. b Three deformation areas of the Tizicao landslide. The red dashed line denotes the boundary of the deformation area. The red flag denotes the location of the 24 fixed non-prism monitoring points (T1−T24). c Rear wall. d Rockslide area, flow area, and accumulated debris.**

**3. Methods section only presents numerical 2D and 3D simulations but not yet describing monitoring instruments and laboratory tests (if you use parameters from the tests for the numerical simulation). Please confirm.**

**Response 3:** Thanks for your suggestion. The monitoring instruments were not described in the manuscript because the monitoring instruments and the monitoring data were given in the reference Zhou et al. (2022), and we referenced this literature. However, we added the descriptions of the monitoring work including monitoring instruments and monitoring points in "2. Study site". Please see the text marked in red in revised manuscript. Thanks.

We took the rock samples from the sliding body, sliding bed, and sliding surface of the landslide to conduct the geotechnical tests. Rock density was obtained by using wax-sealing method. Young's modulus, poisson's ratio, internal friction angle, and cohesion of rocks were collected from the triaxial test. Tensile strength were obtained from the Brazilian test. The obtained rock parameters (Table 1) were used for the numerical simulation. We have added the descriptions of the laboratory tests in the section of "3.2 3D stability simulations" in the revised manuscript. Please see the text marked in red for details. Thanks.

**4. Page 5 Lines 34-35, "The simulation parameters of the sliding body, sliding bed, and sliding surface in the model were obtained through indoor geotechnical tests, shown in Table 1". What kind of geotechnical tests? Please describe.**

**Response 4:** Thanks for your suggestion. In detail, rock density was obtained by using wax-sealing method. Young's modulus, poisson's ratio, internal friction angle, and cohesion of rocks were collected from the triaxial test. Tensile strength were obtained from the Brazilian test. The obtained rock parameters (Table 1) were used for the numerical simulation. We have added the descriptions of the laboratory tests in the section of "3.2 3D stability simulations" in the revised manuscript. Thank you.

**5. Page 6 Lines 4-5, the parameters of tensile strength, shear stiffness, and normal stiffness of the sliding surface for the JM model are assumed or based on the laboratory experiment. Please describe what kind of experiment.**

**Response 5:** As for the JM model, the landslide was simulated by assigning equivalent shear strength parameters to the contact surface model (S3), as shown in Fig. 9b. Then rock bridge

and sliding surface were assumed as the homogeneous contact surface, so the parameters of the assumed contact surface were not set as the same parameters of the rock bridge or sliding surface. Then we took the average value of the tensile strength of the rock bridge and sliding surface, and the real tensile strength of the rock bridge and sliding surface were obtained from the Brazilian test.

As for the shear stiffness and normal stiffness of the sliding surface in the JM model, there was no direct reference yet, so we referenced the "FLAC3D6.0 Theory and Background". A good rule-of-thumb is that and be set to ten times the equivalent stiffness of the stiffest neighboring zone. The apparent stiffness of a zone in the normal direction is,

$$\max[\frac{K+\frac{4}{3}G}{\Delta z_{\min}}] \tag{1}$$

Where, $K$ & $G$ are the bulk and shear moduli, respectively; $\triangle z_{\min}$ is the smallest width of an adjoining zone in the normal direction.

The [ ] notation indicates that the maximum value over all zones adjacent to the interface is to be used.

For the soils in sliding surface, their Poisson's ratio and Young's modulus are 0.3 and 3.51 GPa, respectively, then their bulk moduli and shear moduli are 2.93 GPa and 0.51 GPa, respectively. For the mesh model of the landslide, its smallest width of an adjoining zone in the normal direction is 2 m. We substituted the bulk moduli, shear moduli, and the smallest width of an adjoining zone into above equation (1), and the normal stiffness of the sliding surface was determined to be 1801.63 MPa/m. So we assigned the value of 1800 MPa/m to the shear stiffness and normal stiffness of the sliding surface.

**6. Page 6 Lines 8-10, the strength parameters and stiffness coefficients of the sliding surface in the CSM-HSP and the IRMM were set the same. Could you please explain why?**

Response 6: In Fig. 9a, the sliding surface (S2) was replaced by the contact surface for the IRMM, also the sliding surface (S5) was simulated by the contact surface for the CSM-HSP in Fig. 9c. Then the sliding surface model and its simulation parameters were the same in the FLAC3D program. For the CSM-HSP and the IRMM, only the rock bridge model was

different.

**7. Please consider the bar scale in the map of Sichuan Province in Figure 1.**

**Response 7:** Thank you for your suggestion. We have added the bar scale in Figure 1 in revised manuscript.

[Figure]

**Figure 1: Location of the Tizicao landslide in Sichuan Province, southwest China. Source: ©Google Earth Pro 2021.**

**8. Correct me if Figure 2b is an aerial/oblique view of the landslide? is it necessary to put a bar scale? I prefer to provide a bar scale for Figure 2a as it is an orthogonal image. Please confirm.**

**Response 8:** Figure 2b is an oblique view of the landslide. It is necessary to put a bar scale and we added it in Figure 2b. Also we added a bar scale in Figure 2a below.

[Figure]

**Figure 2: Overall perspective of the site area of the Tizicao landslide (after Zhou et al., 2022). a An orthoimage of the landslide site area taken on November 24, 2020, with a resolution of 3840 ✕ 2160. b Three deformation areas of the Tizicao landslide. The red dashed line denotes the boundary of the deformation area. The red flag denotes the locaiton of the 24 fixed non-prism mointoring poingts (T1−T24). c Rear wall. d Rockslide area, flow area, and accumulated debris.**

**9. In Figure 4a, section A-A' the L07 should be L04, while in Figure 4b L11 not indicated in section B-B'. Then Figure 4c the monitoring points of T1 and T2 nearby L03 are not present. Please revised.**

**Response 9:** Thank you for your so careful reading. In fact, the L07 is not the L04 in section A-A' in Figure 4a, and we carelessly forgot to mark the L04, then we added the L04 in Figure 4a. The L11 was not indicated in section B-B' and we revised the Figure 4b. In Figure 4c, we added the monitoring points of T1 and T2 nearby L03. Please see the revised Figure 4 below in detail. Thanks.

[Figure]

**Figure 4: Engineering-geotechnical sections of the Tizicao landslide. a Section A-A'. b Section B-B'. c Section C-C'. d Section D-D' (after Zhou et al., 2022).**

**10. Location of borehole zk20 provided in Figure 5 is not shown on the map in Figure 3. Please add and clarify.**

**Response 10:** Thank you for your so careful reading. We added the location of borehole zk20 in Figure 3a below.

[Figure]

**Figure 3: The topographic plan, isoline map of surface displacement, and displacement monitoring curves of the Tizicao landslide. a** Topographic plan of the deformation areas, the crack distribution, and the locations of engineering-geotechnical sections (after Zhou et al., 2022). **b** Isoline map of the surface displacement of the Tizicao landslide from August 13, 2017 to January 25, 2018 (Zhou et al., 2022); **c** Displacement monitoring curves of the landslide surface (from August 13, 2017 to January 25, 2018).

**11. Locations of zk08 in Figure 7a and exposed phyllites in Figures 7b and 7c are also not pointed on the map. Please add and clarify.**

**Response 11:** Thank you for your so careful reading. We added the location of borehole zk08 in Figure 3a. The exposed phyllites in Figures 7b and 7c were also pointed in the Figure 3a below. Please see the revised Figure 3 in detail. Thanks.

[Figure]

**Figure 3: The topographic plan, isoline map of surface displacement, and displacement monitoring curves of the Tizicao landslide. a** Topographic plan of the deformation areas, the crack distribution, and the locations of engineering-geotechnical sections (after Zhou et al., 2022). **b** Isoline map of the surface displacement of the Tizicao landslide from August 13, 2017 to January 25, 2018 (Zhou et al., 2022); **c** Displacement monitoring curves of the landslide surface (from August 13, 2017 to January 25, 2018).

**12. Please provide a full description of each abbreviation in the caption of Figure 9.**

**Response 12:** Thank you for your so careful reading. The full description of IRMM is intact rock mass model, and the full description of JM is Jennings model, and the full description of CSM-HSP is contact surface model with high strength parameters. We added the full description of each abbreviation in the caption of Figure 9.

[Figure]

Block A: sliding body
Block B: sliding bed
S1: rock bridge
S2: sliding surface

Block A: sliding body
Block B: sliding bed
S3: sliding surface with the
equivalent strength parameters

Block A: sliding body
Block B: sliding bed
S4: rock bridge replaced by sliding
surface with high strength parameters
S5: sliding surface

**Figure 9: Three rock bridge models used in the FLAC3D program. a Intact rock mass model (IRMM). b Jennings model (JM). c contact surface model with high strength parameters (CSM-HSP).**

**13. It is not clear whether the red zone of the sliding surface in Figure 10 is shown in the mesh model. Please clarify.**

**Response 13:** The red zone represented the sliding surface in the mesh mode in Figure 10. Because of the opacity of sliding body mesh, we can't see the full view of the red zone of the sliding surface. Then we added the mesh of the sliding surface in Fig. 10 below.

[Figure]

**Figure 10: The mesh model and geometry of the Tizicao landslide.**

**14. Cohesion between A-A', B-B' and C-C', D-D' in Figure 11 for 2D simulation are very different significantly, why? Check also its relation with simulation parameters in Table 1. Please discuss.**

**Response 14:** The differences of cohesion between A-A', B-B' and C-C', D-D' in Figure 11 for 2D simulation was because of the locking ratio $k_L$ (the ratio of the surface area of the rock bridge to the total sliding surface area). For the section A-A' and B-B', the locking ratio $k_L$ = 0, i. e., there was no rock bridge in these sections. Then the shear strength parameters of the sliding surface can be set as the same with those in Table 1. However, for the section C-C' and D-D', the locking ratios $k_L$ = 0.23, and 0.26, respectively. According to the JM, the slope stability is calculated by assigning the equivalent shear strength corresponding to different penetration rates to the potential sliding surface. The equivalent shear strength parameters can be calculated as follows:

$$c_{eq} = (1-k)c_r + kc_j \tag{1}$$

$$\tan\varphi_{eq} = (1-k)\tan\varphi_r + k\tan\varphi_j \tag{2}$$

where, $c_{eq}$ and $\varphi_{eq}$ are the equivalent cohesion and the equivalent friction angle, respectively; $\varphi_r$ and $\varphi_j$ represent the friction angle of intact rock and joints, respectively, and $c_r$ and $c_j$ are the cohesion of intact rock and joints, respectively.

Considering that co-planar joints are separated by the intact rock bridge, the relative quantity of intact rocks along the sliding surface can be expressed by the ratio $k$, which is defined as (Jennings, 1970):

$$k = \frac{\sum A_j}{\sum A_j + \sum A_r} = 1 - k_L \tag{3}$$

where, $\sum A_j$ denotes the surface area of joints, $\sum A_r$ is the surface area of the rock bridge, and $k_L$ is the locking ratio (the ratio of the surface area of the rock bridge to the total sliding surface area).

So when we calculated the 2D Fos, we used the equivalent cohesion $c_{eq}$ and the equivalent friction angle $\varphi_{eq}$. The equivalent shear strength parameters were obtained from Eq. (1) and Eq. (2) by using shear strength parameters of the rock bridge and the sliding surface in Table 1 as well as the different locking ratios in Figures 11c-d. Then the equivalent shear strength parameters were calculated and shown in Figures 11c-d.

[Figure]

**Figure 11: 2D Fos of different sections. a Section A-A'. b Section B-B'. c Section C-C'. d Section D-D'.**

**15. Contours of shear displacement and intense deformation zone in Figure 12 seems not in line with the isoline map of surface displacement in Figure 3b. Perhaps unclear explanation in the text that I could not catch. Could you emphasize the above results?**

**Response 15:** For the unclear explanation of the deformation characteristics between the simulation results in Fig. 12 and the isoline map of surface displacement in Fig. 3b. We added the description of the characteristics of the surface displacement in Fig. 3b,which was described as follows:

As shown in the isoline map of surface displacement (Fig. 3b), a sliding event occurred in a general northeast direction (closer to the north) from August 13, 2017 to January 25, 2018. In this event, the maximum surface displacement (1210 mm) occurred at the northern toe, which coincided with the location where the front collapsed (Fig. 2d). The landslide's rear and middle parts showed similar surface displacement of 150–300 mm in the sliding event, indicating that they slid as a whole. The minimum surface displacement of 30–150 mm occurred in the southern area of the slope toe throughout the whole sliding event. Therefore, the southern area serves as the anti-sliding area of the whole landslide.

As shown in Fig. 12a, the total displacement contours of the sliding body show significantly different deformation zones, namely the intense deformation zone from the rear to the north side wall of the landslide, the moderate deformation zone from the middle part of the landslide to the northern part of the landslide front, and the slight deformation zone in the middle and southern parts of the landslide front. The maximum displacement of the sliding body is 10.69 m at the landslide's rear (Fig. 12a), which agrees with the width of crack L07 (Fig. 13). Fig. 12a shows that the Tizicao landslide tends to slide northeastward generally owing to the sliding resistance of the locking segment. This tendency is consistent with the crack distribution (Fig. 3a) and the isoline map of surface displacement (Fig. 3b). Figs. 12a and 3a show different displacement values because the monitoring data obtained from August 13, 2017 to January 25, 2018 (after the large deformation in July 2017) were not the complete deformation data of the landslide. In contrast, Figs. 12a and 3a reflect the same deformation tendency.

**16. Font remark in Figure 13 is not visible, please change its color.**

**Response 16:** Thanks for your suggestion. We changed the yellow color into black color in Figure 13.

[Figure]

**Figure 13: Crack L07 at the rear of the landslide. The width of crack L07 is 10.5 m in the direction of the landslide.**

**17. The locking segment needs to be described further, is it generated due to the movement characteristic of a landslide? Or originally due to the lithological or geological conditions? Or are there any factors or settings that cause this feature? Please discuss.**

**Response 17:** Thanks for your suggestion. In the section of "2. Study site", we described the location, area, lithology, RQD values with different depth, and surface deformation characteristics of the locking segment. However, we did not describe the origin cause of the locking segment.

In fact, Zhou et al. (2022) preliminarily discussed the cause. The locking mass of the Tizicao landslide developed on the convex bank, while the non-locking mass developed on the concave bank, indicating that the locking mass was directly related to that of the S-shaped river valley. From a geomorphological point of view, landslides rarely occur on the convex banks, but occur more frequently on concave banks. From a topographical point of view, the convex slope is more stable than the concave slope under the same conditions. Noticeably, the concave and convex bank parts of the S-shaped valley under the Tizicao landslide are greatly different in slope and lithology. Therefore, the rock masses on the southern side of the landslide above the convex bank part reserve their integrity and constitute the potential locking segment of the landslide.

We added above descriptions in the section of "2. Study site" to explain the cause of the locking segment.

**RC2: 'Comment on egusphere-2023-28', Anonymous Referee #2, 16 Apr 2023**

**English**

**1. Firstly, the English language requires revision as it includes numerous errors and repetitions, and the form requires adjustment as it can be unclear and confusing at times.**

**Response 1:** Thank you for your valuable and thoughtful comments. We have asked a professional proofreading company to improve the writing of this manuscript, and a native English speaker has carefully checked and polished the language of this manuscript.

**Geomechanical characterization**

**2. Another issue is the geomechanical characterization. How was it conducted, and what tests were performed to acquire the mechanical parameters? For instance, the sentence "the equivalent shear strength parameters were determined based on penetration rates" on page 6, line 15, needs explanation about the method used to determine these parameters.**

**Response 2:** Thank you for your suggestions. As for the geomechanical characterization, we took the rock samples from the sliding body, sliding bed, and sliding surface of the landslide to conduct the geotechnical tests. Rock density was obtained by using wax-sealing method. Young's modulus, poisson's ratio, internal friction angle, and cohesion of rocks were collected from the triaxial test. Tensile strength were obtained from the Brazilian test. The obtained rock parameters (Table 1) were used for the numerical simulation. We have added the descriptions of the laboratory tests in the section of "3.2 3D stability simulations" in the revised manuscript. Plesase see the text marked in red for details. Thanks.

For the sentence "the equivalent shear strength parameters were determined based on penetration rates" on page 6, line 15. According to the JM, the slope stability is calculated by assigning the equivalent shear strength corresponding to different penetration rates to the potential sliding surface. The equivalent shear strength parameters can be calculated as follows:

$$c_{eq} = (1-k)c_r + kc_j \tag{1}$$

$$\tan \varphi_{eq} = (1-k)\tan \varphi_r + k\tan \varphi_j \tag{2}$$

where, $c_{eq}$ and $\varphi_{eq}$ are the equivalent cohesion and the equivalent friction angle, respectively; $\varphi_r$ and $\varphi_j$ represent the friction angle of intact rock and joints, respectively, and $c_r$ and $c_j$ are the cohesion of intact rock and joints, respectively.

Considering that co-planar joints are separated by the intact rock bridge, the relative quantity of intact rocks along the sliding surface can be expressed by the ratio $k$, which is defined as (Jennings, 1970):

$$k = \frac{\sum A_j}{\sum A_j + \sum A_r} = 1 - k_L \tag{3}$$

where, $\sum A_j$ denotes the surface area of joints, $\sum A_r$ is the surface area of the rock bridge, and $k_L$ is the locking ratio (the ratio of the surface area of the rock bridge to the total sliding surface area).

So the equivalent shear strength parameters were obtained from Eq. (1) and Eq. (2) by using shear strength parameters of the rock bridge and the sliding surface in Table 1 as well as the different locking ratios in Figures 11c-d. Then the equivalent shear strength parameters were calculated and shown in Figures 11c-d.

**Numerical Modeling**

**3. The authors don't explain how the numerical analyses were set up in Flac3D. The paper should clarify how the IRMM, JM, and CSP-HSP constitutive models were implemented in the Flac3D code. The reviewer is wondering if Perfectly Plastic Mohr-Coulomb model was modified.**

Response 3: We are sorry for the lack of clarification how these model were implemented in the FLAC3D code. For the IRMM, JM, and CSP-HSP models, they are the simplified models of landslides with rock bridges. We use different elements (tetrahedral elements, contact surface elements) in FLAC3D code to replace rock bridges and sliding surface. The details are described as follows,

For the IRMM model, as shown in Fig. 9a, in the simulation of a landslide with a locking segment, a rock bridge (S1) is an intact rock mass, and it is simulated by tetrahedral elements in FLAC3D program. The contact surface model in FLAC3D program is used to simulate the sliding surface (S2). The sliding body (Block A) and the sliding bed (Block B) are linked with a continuous rock bridge (S1).

[Figure]

Block A: sliding body
Block B: sliding bed
S1: rock bridge
S2: sliding surface

Block A: sliding body
Block B: sliding bed
S3: sliding surface with the equivalent strength parameters

Block A: sliding body
Block B: sliding bed
S4: rock bridge replaced by sliding surface with high strength parameters
S5: sliding surface

**Figure 9: Three rock bridge models used in the FLAC3D program. a Intact rock mass model (IRMM). b Jennings model (JM). c contact surface model with high strength parameters (CSM-HSP).**

For the JM model (Jennings, 1970), the limit equilibrium method is initially employed to calculate the 2D stability of rock slopes with discontinuous joints. In detail, the slope stability is calculated by assigning the equivalent shear strength corresponding to different penetration rates to the potential sliding surface. Herein we introduced the Jennings model into the FLAC3D program. Then we simulated the 3D stability of the whole landslide (rock bridge, sliding surface) by assigning equivalent shear strength parameters to the contact surface model (S3), as shown in Fig. 9b.

As shown in Fig. 9c, two contact surface models with high and low strength parameters each were used to simulate the rock bridge (S4) and sliding surface (S5), respectively. The strength parameters of an intact rock mass are adopted for the rock bridge. In addition, shear stiffness and normal stiffness higher than those of the sliding surface are required in the CSM-HSP model to simulate the real resistance characteristics of the rock bridge.

According to above descriptions, we did not establish new constitutive models implemented in the Flac3D code. We just simplified the landslide models, and provided different methods to simulate the landslide with rock bridges in FLAC3D program using different mesh elements. In the simulation of the landslide with IRMM, JM, and CSP-HSP models, all the mesh elements yields Plastic Mohr-Coulomb model, so the Perfectly Plastic Mohr-Coulomb model was not modified in this manuscript.

**4. The authors mention Geostudio as a software utilized for 2D calculations, however this sentence doesn't explain the 2D analysis method because Geostudio is a software suite which includes different products so the review is wondering if the numerical simulations were carried out with Limit Equilibrium Method (LEM) of the Slope/W**

**software. The paper should explain how the JM model was introduced into the Bishop algorithm in LEM modelling.**

**Response 4:** Thank you for your comments. We used the SLOPE/W module of the program GeoStudio 2012 to conduct the 2D stability analysis of the Tizicao landslide. The SLOPE/W module of the program GeoStudio 2012 was used to calculate 2D stability of slope/landslide (Chen et al., 2020; Jafri et al., 2020). In the SLOPE/W module of the program GeoStudio 2012, there are many Limit Equilibrium Methods to calculate the 2D Fos. We chose one Limit Equilibrium Method, i. e., Bishop's algorithm method for 2D calculations. Meanwhile, the JM model was introduced into Bishop's algorithm of the GeoStudio program. In the JM model, we only input the equivalent shear strength determined based on penetration rates by using Eq. (1) and Eq. (2) for the sliding surface, and the slope stability can be calculated. For the sections A-A', B-B', C-C', and D-D', according to the site survey, the locking ratios $k_L$ are 0, 0, 0.23, and 0.26, respectively. The equivalent shear strength were calculated in Fig. 11, and the calculated 2D stability factors were shown in Table 2 and Fig. 11.

[Figure]

**Figure 11: 2D Fos for different sections. a Section A-A'. b Section B-B'. c Section C-C'. d Section D-D'.**

**5. The paper should provide better specification of the comparisons between the four 2D surfaces and the hypothesized 3D surface.**

**Response 5:** Thank you for your suggestions. For the 2D sliding surfaces, they were deduced according to the depth of the sliding zone soil obtained by drilling, as shown in Fig. 4. Meanwhile, the geometric size and shape of the 3D sliding surface were deduced according to the four 2D sliding surfaces and the outline of the landslide. So the 3D sliding surface was the deduction result of the four 2D sliding surfaces.

As for the comparisons between the 2D and 3D stability of the landslide, we added the detailed comparisons between the four 2D stability and the 3D stability, which was described as follows,

For the landslide sections (sections A-A', B-B') with severe deformation, 2D Fos was lower than 3D Fos. However, for the landslide sections (sections C-C', D-D') with slight deformation, 2D Fos were greater than 3D Fos significantly, especially for the landslide sections with the locking segment. 2D stability analysis is relatively conservative (Li et al., 2010; Park et al., 2017), i. e., 2D Fos is usually lower than the 3D Fos. Nonetheless, for the landslide sections with rock bridges, 2D Fos may exceed the 3D Fos (Table 2). Then the overall stability of the landslide with rock bridges should be assessed by the 3D Fos, as the 2D Fos only represent the local stability.

**Monitoring**

**6. Lastly, while the paper often discusses monitoring of deformations, it fails to explain how the monitoring was conducted and the tools used to obtain the data.**

**Response 6:** Thank you for your so careful reading. In fact, we conducted the monitoring work of the landslide for several years. The monitoring work was conducted by the Leica Monitoring Total Station (TM50), and it was set up on the slope opposite the landslide. In the landslide body, twenty-four fixed non-prism monitoring points (T1−T24) were deployed to mainly monitor the surface displacement from June 1, 2017, to October 2, 2017, as shown in Figure 8 in reference Zhou et al. (2022). They almost covered the entire landslide body. These raw data of the surface displacement were processed using the measurement adjustment software DDM to obtain their deformation amount and deformation rate. We added the descriptions of the monitoring work in "2. Study site". Please see the text marked

in red in revised manuscript. Thanks.

---

## Author Response (AR3)

**Comments response on "Simulation analysis of 3D stability of a landslide with a locking segment: A case study of the Tizicao landslide in Maoxian County, Southwest China"**

**Report #1      Submitted on 20 Jul 2023**

**Anonymous referee #2**

Globally, the authors have responded to the questions raised and with the additions, it becomes clearer to understand the work done. However, some concerns still remain regarding the structure and the English language usage. Additionally, the overall organization of the paper may need to be reviewed.

**1.Introduction: The motivation and importance of the study could be better focused and made more apparent.**

Response 1: Thank you for your valuable and thoughtful comments. In the "Introduction" section, we first described the potential catastrophic effects of the landslide with a locking segment and the critical role of 3D stability analysis in assessing and predicting the overall stability of the landslide with a locking segment. Then, we reviewed literature about 3D stability methods and pointed out the disadvantages of these methods in the analysis of landslides with rock bridges. Furthermore, the parameters of rock bridges, such as length, penetration rate, strength parameters, joint strength parameters, relative positions (direction, coplane, or non-coplane), and shape of rock bridges, were determined since they should be considered in the 3D stability for the landslide with a locking segment. Lastly, three objectives of this work are proposed as follows:

(1) The first objective is to present an improved rock bridge model and to simulate the 3D stability and deformation behaviors of the Tizicao landslide using the model.

(2) The second objective is to explore the advantages and disadvantages of the three rock bridge models in the simulation of the 3D stability of landslides with a locking segment.

(3) The last objective is to explore the laws of the 3D Fos varying with the locking ratios and strength parameters of the locking masses and the sliding surface.

**2.Methodology: The methodology section appears to be somewhat confusing. It would be better to organize all the methods used in a more orderly manner.**

Response 2: We are sorry for the confusion of the methodology section. We added one paragraph to describe the process and operation of the methodology as follows:

The 3D stability of the Tizicao landslide was simulated using the FLAC3D program. First, we introduced three rock bridge models, namely IRMM, JM, and CSM-HSP, into the FLAC-3D program and determined the simulation elements and their characteristic parameters. According to the site survey of the Tizicao landslide, a 3D mesh model composed of a sliding bed, a sliding body, and a sliding surface was established. Then, we simulated the 3D stability of the Tizicao landslide using the three rock bridge models. Lastly, to compare the differences between the 2D and 3D stability of the Tizicao landslide, this study analyzed the 2D stability of four sections of the landslide using the SLOPE/W module of the GeoStudio 2012 program.

Please see the revised manuscript in detail. Thanks.

**3.Results and discussion: This section should be the most interesting and crucial part of the paper, but it seems underdeveloped. A critical discussion of the results, their implications, limitations, and uncertainties is lacking.**

Response 3: Thanks for your valuable and thoughtful comments. In the "Results and discussion" section, we further discussed the effects of the locking ratio on 3D stability. We added descriptions of the influence of the positions of locking masses on the 3D stability of landslides with rock bridges. Additionally, we pointed out that this study did not explore the effects of the curvature of the sliding surface on 3D stability, which will be discussed in future research. Besides, we added the statistics of rock bridge content obtained by previous researchers to explain the dramatic failure in the case of very low rock bridge content. Please see the revised manuscript in detail. Thank you.

**4.Novelty/Originality: The novelty of the study does not appear to be adequately demonstrated.**

Response 4: The novelty of this study is as follows:

(1) Three rock bridge models, i.e., IRMM, JM, and CSM-HSP, are established to simulate the 3D stability and deformation behaviors of the landslide with a locking segment.

(2) The advantages and disadvantages of the three rock bridge models are provided to

guide the 3D stability of the landslide with a locking segment.

(3) This study presents the laws of the 3D Fos varying with the locking ratios and strength parameters of the locking masses and the sliding surface.

In addition, we have asked a professional proofreading company to improve the English language usage of the modifications, which were carefully checked and polished by a native English speaker.

**Report #2    Submitted on 03 Aug 2023**

**Referee #1: Hendy Setiawan, hendy.setiawan@ugm.ac.id**

There is no further suggestions for revision.